

# A surface energy and mass balance model for simulations over multiple glacial cycles

Andreas Born[1,2,3,4], Michael A. Imhof[3,5], and Thomas F. Stocker[3,4]

[1]Department of Earth Science, University of Bergen
[2]Bjerknes Centre for Climate Research, University of Bergen
[3]Climate and Environmental Physics, University of Bern, Bern, Switzerland
[4]Oeschger Centre for Climate Change Research, University of Bern, Bern, Switzerland
[5]Laboratory of Hydraulics, Hydrology and Glaciology, ETH Zürich, Zürich, Switzerland

*Correspondence to:* Andreas Born (andreas.born@uib.no)

**Abstract.** A comprehensive understanding of the state and dynamics of the land cryosphere and associated sea level rise is not possible without taking into consideration the intrinsic time scales of the continental ice sheets. At the same time, the ice sheet mass balance is the result of seasonal variations in the meteorological conditions. Simulations of the coupled climate-ice sheet system thus face the dilemma of skillfully resolving short-lived phenomena, while also being computationally fast

enough to run over tens of thousands of years. Further complications arise from the fact that the mass balance is a small residual of various contributions that individually are much larger, and that even a marginal bias will develop into an erroneous solution over the long integration time and when amplified by strong positive feedback mechanisms. As a possible solution, we present the BErgen Snow SImulator (BESSI), a surface energy and mass balance model that achieves computational efficiency while simulating all surface and internal fluxes of heat and mass explicitly and based on physical first principles. In its current

configuration it covers most land areas of the Northern Hemisphere. Two large ensembles of simulations are investigated, one to calibrate the model and another one to assess its sensitivity to variations in air temperature.

## 1 Introduction

Polar ice sheets are an important component of the earth system with far-reaching impacts on global climate. They lock vast amounts of water on land and thus lead to a considerably lower sea level (Waelbroeck et al., 2002). Furthermore, ice sheets

radically change the topography and therefore have a profound influence on the atmospheric circulation (Li and Battisti, 2008; Liakka et al., 2016). There is thus a great interest in simulating the dynamic changes of ice sheets over time. One of the fundamental issues that these efforts need to address is the large discrepancy between the typical response times of the flow of ice, on the order of millennia for ice sheets of continental size, and the diurnal to seasonal time scale on which relevant processes at the snow surface change the energy and mass balance. The long-term mass balance may be notably reduced when subject to

high-frequency variability in the forcing because of its highly nonlinear response to deviations from the climatological average (Mikkelsen et al., 2018). This effect is exacerbated by the fact that even initially very small changes in the surface mass balance





accelerate over time leading to the complete loss of ice in certain regions (Born and Nisancioglu, 2012). Moreover, the loss of ice causes profound changes in the local climate which feed back into the energy and mass balances (Merz et al., 2014a, b).

This complexity calls for the bi-directional coupling of ice sheet and climate models. However, the large difference in characteristic time scales and therefore simulation time means that modern comprehensive climate models are prohibitively

expensive to run for periods that are relevant for ice sheets. While the interpolation of time slice simulations is a viable alternative (e.g., Abe-Ouchi et al., 2013), climate models of reduced complexity provide a solution that foregoes the problematic temporal interpolation and achieves a more direct coupling (Bonelli et al., 2009; Robinson et al., 2011). The problem with this approach is that the necessary compromises in complexity require that these model employ only rudimentary land surface schemes, do not simulate atmosphere dynamics and parameterize many processes that are potentially important for the ice

sheet energy and mass balances such as the cloud cover.

In this study we present the BErgen Snow SImulator (BESSI), an efficient surface energy and mass balance model designed for use with coarse resolution earth system models. The main development objectives were to include all relevant physical mechanisms with a reasonable degree of realism, albeit with reduced complexity to not rely on forcing fields that would not be available from the simplified climate models, and to achieve a computational speed that matches that of the climate models for

the capability to run simulations over multiple glacial cycles (Ritz et al., 2011).

Existing surface mass balance (SMB) models have a wide spread of complexity, ranging from empirical temperature index models, also known as positive degree day (PDD) models (Reeh, 1991; Ohmura, 2001), to comprehensive physical energy balance models that resolve the snow surface at millimeter scale and with an evolving snow grain microstructure (Bartelt and Lehning, 2002). While empirical models that calculate the mass balance from fields of air temperature and precipitation alone

perform adequately in specific regions and stable climate (Hock, 2003), their results become unreliable for large regions with spatially heterogenic and variable climatic conditions (Bougamont et al., 2007; van de Berg et al., 2011; Robinson and Goelzer, 2014; Gabbi et al., 2014; Plach et al., 2018). More sophisticated models can potentially simulate the SMB with higher fidelity by explicitly calculating the important effects of short- and longwave radiation, vertical heat transport in the snow, aging and albedo changes at the snow surface, the densification of the snow with depth, and the retention of liquid water (Greuell, 1992;

Greuell and Konzelmann, 1994; Bougamont et al., 2005; Pellicciotti et al., 2005; Reijmer and Hock, 2008; Vionnet et al., 2012; Reijmer et al., 2012; Steger et al., 2017). However, the accurate simulation of these processes demands far greater detail on meteorological conditions than is available from coarse climate models or reconstructions of paleoclimate, and their high level of detail makes these models too slow for long-term simulations.

This situation has motivated the development of efficient SMB models that aim to balance the representation of relevant

physics with numerical efficiency and manageable prerequisites for the input data. A common simplification is the reduction to a single snow layer and the parameterization of processes shorter than a single day (Robinson et al., 2010; Krapp et al., 2017). BESSI simulates the snow and firn on 15 layers that reach the maximum depth of seasonal temperature variability, approximately 15 m. The full energy balance is calculated at the snow surface and the diffusion of heat below that. The percolation of liquid water, either from melting or from rain, is formulated explicitly. Both overburden pressure and refreezing

of meltwater contribute to the densification. Input fields are the shortwave radiation, air temperature and precipitation. The



numerical performance for simulations of the Northern Hemisphere with 40 km resolution is 300 model years per hour on a single modern CPU.

Section 2 describes the model, how physical processes are simulated as well as their temporal and spatial discretization. This section also contains information on the climatic forcing and the model domain. Section 3 discusses the conservation

of mass and energy in the model and presents simulations designed to constrain the uncertainty of poorly known parameters using modern observations. Section 4 presents some key characteristics of BESSI, followed by applying the calibrated model to assess the trend and nonlinearity of the SMB with respect to anomalous temperatures in section 5. We conclude our results in the final section 6.

## 2   Model description

### 2.1   Model domain and discretization

The model domain is a square centered on the North Pole, discretized as a 40 km equidistant grid based on a stereographic projection (Fig. 1). Each horizontal axis has 313 grid cells. This grid is identical to the one used in the fast ice sheet model IceBern2D, into which BESSI is integrated as a subroutine (Neff et al., 2016). However, ice dynamics are disabled in all simulations shown here. Bedrock data are taken from the ETOPO1 dataset (Amante and Eakins, 2009), bi-linearly interpolated

onto the new grid.

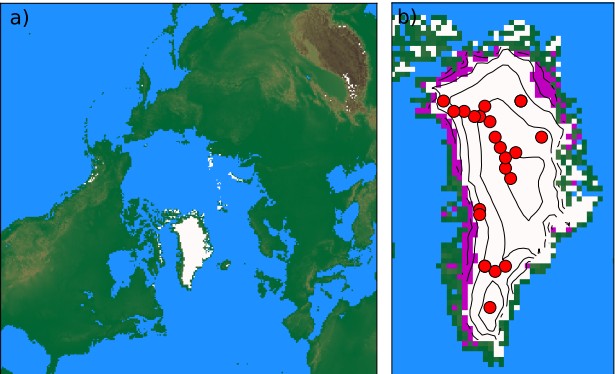

**Figure 1.** a) Model domain used in this study and topography based on the ETOPO1 dataset (Amante and Eakins, 2009). White areas illustrate regions with a net positive mass balance in simulation $\text{BEST}_T$ (Sec. 3.2). b) in addition to a) magenta colors show net negative mass balance and contour lines show elevation over ice, spacing 500m. The red dots indicate locations of firn temperature measurements (Tab. 3).

The vertical dimension is implemented on a mass-following dynamical grid with a given maximum number of vertical layers $n_{\text{layers}}$. This number can in principle be chosen freely but is constant at 15 throughout this study. However, not all layers are filled at all locations at all time steps. Precipitation fills the uppermost box until it reaches the threshold value of $m_{\text{max}} = 500$




kg m$^{-2}$. At this point, the box is split in two such that the lower part contains $m_{\mathrm{split}} = 300$ kg m$^{-2}$ of snow and the upper one the remaining mass. To accommodate the new subsurface grid box, the entire snow column below the former surface is shifted one grid box downward (Fig. 2a). After the split, temperature and density do not change, and the liquid water is distributed using the same ratios as for the snow mass. In case all $n_{\mathrm{layers}}$ boxes were full before the split, the two lowest boxes are merged

into one.

Similarly, if the snow mass in the surface box sinks below $m_{\mathrm{min}} = 100$ kg m$^{-2}$, it is combined with the second box, if there is one (Fig. 2b). Should the combined weight be at more than twice as heavy as $m_{\mathrm{split}}$, the mass is transferred only partially so that the surface box does not become more heavy than $m_{\mathrm{split}}$.

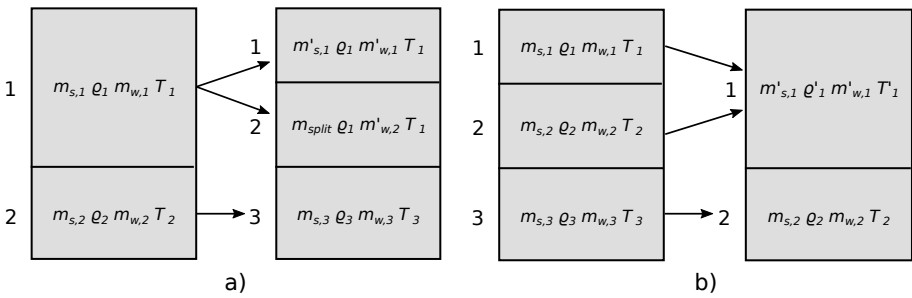

**Figure 2.** (a) Splitting of a box that exceeds the maximum snow mass $m_{\mathrm{max}}$ and (b) Merging of two boxes in case the mass of the uppermost box falls below $m_{\mathrm{min}}$.

Thus, of the four variables that are simulated prognostically on the grid, snow mass $m_{\mathrm{s}}$, liquid water $m_{\mathrm{w}}$, density of snow

$\rho_{\mathrm{s}}$, and snow temperature $T_{\mathrm{s}}$, liquid water is the only one that is regularly exchanged between boxes, disregarding the relatively seldom merging or splitting of boxes (Fig. 3). This simplifies the model formulation, because the advection equation does not have to be solved and it avoids spurious fluxes of heat and other tracers due to numerical diffusion. For the same reason boxes with depth index 2 to 14 can only contain masses above $m_{\mathrm{split}} = 300$ kg m$^{-2}$ where liquid water refreezes. A conceptually similar vertical grid is used in the isochronal ice sheet model of Born (2016). Horizontal interactions between columns are not

simulated.

The surface box exchanges energy with the atmosphere ($Q_{SF}$) in the form of sensible, latent, and radiative heat. It receives mass as either rain ($\Delta m_r$) or snow ($\Delta m_{\mathrm{s}}$). Temperature diffuses through the layers ($Q_{TD}$), and the snow pack densifies with time and rising overburden pressure. Meltwater and rain percolate ($\Delta m_{\mathrm{w}}$) and may freeze again, which leads to internal accumulation. Water leaving the lowest grid cell is treated as runoff. Lists of all model variables and constants are found in

Tables 1 and 2, respectively.





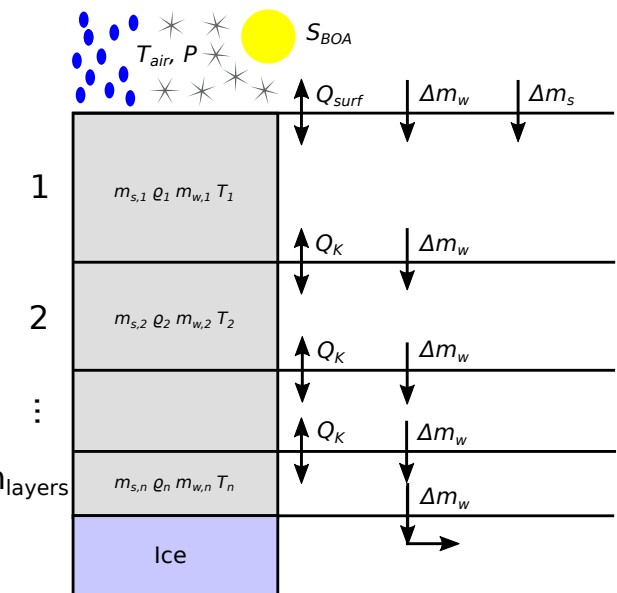

**Figure 3.** Setup of snow layers with fluxes of energy and mass and snow quantities in one snow column.

**Table 1.** Table with all symbols used in the model

| Quantity | Symbol | Units |
|---|---|---|
| Snow mass per area | $m_{\mathrm{s}}$ | kg m$^{-2}$ |
| Liquid water mass per area | $m_{\mathrm{w}}$ | kg m$^{-2}$ |
| Density of snow | $\rho_{\mathrm{s}}$ | kg m$^{-3}$ |
| Temperature of snow | $T_{\mathrm{s}}$ | K |
| Temperature of air | $T_{\mathrm{air}}$ | K |
| Precipitation | $P$ | m s$^{-1}$ |
| Solar radiation at the bottom of the atmosphere | $S_{\mathrm{BOA}}$ | W m$^{-2}$ |
| Net shortwave radiation | $Q_{\mathrm{sw}}$ | W m$^{-2}$ |
| Net longwave radiation | $Q_{\mathrm{lw}}$ | W m$^{-2}$ |
| Net heat flux due to precipitation | $Q_{\mathrm{p}}$ | W m$^{-2}$ |
| Net Sensible heat flux | $Q_{\mathrm{sh}}$ | W m$^{-2}$ |
| Thermal Conductivity of snow | $K$ | W m$^{-1}$ K$^{-1}$ |



**Table 2.** Table of physical constants and model constants. The tuning parameters have their tested values listed in brackets.

| Quantity | Symbol | Value | Units |
|---|---|---|---|
| Density of fresh snow | $\rho_{s,0}$ | 350 | $\mathrm{kg\,m^{-3}}$ |
| Density of ice | $\rho_i$ | 917 | $\mathrm{kg\,m^{-3}}$ |
| Density of water | $\rho_w$ | 1000 | $\mathrm{kg\,m^{-3}}$ |
| Thermal conductivity of ice | $K_i$ | 2.1 (at 0°C) | $\mathrm{W\,m^{-1}\,K^{-1}}$ |
| Heat capacity of ice | $c_i$ | 2110 (at -10°C) | $\mathrm{J\,kg^{-1}\,K^{-1}}$ |
| Heat capacity of water | $c_w$ | 4181 (at 25°C) | $\mathrm{J\,kg^{-1}\,K^{-1}}$ |
| Latent heat of melting | $L_{lh}$ | 334,000 | $\mathrm{J\,kg^{-1}}$ |
| Coefficient for sensible heat flux | $D_{sh}$ | [5, 10, 20] | $\mathrm{W\,m^{-2}\,K^{-1}}$ |
| Albedo of fresh snow | $\alpha_{dry}$ | [0.75, 0.8, 0.85, 0.9] | 1 |
| Albedo of wet snow | $\alpha_{wet}$ | [0.5, 0.6, 0.7] | 1 |
| Albedo of ice | $\alpha_{ice}$ | [0.35] | 1 |
| Emissivity of air | $\epsilon_{air}$ | [0.6, 0.65, 0.7, 0.75, 0.8, 0.85, 0.9] | 1 |
| Emissivity of snow | $\epsilon_{snow}$ | 0.98 | 1 |
| Minimum mass of a box | $m_{min}$ | 100 | $\mathrm{kg\,m^{-2}}$ |
| Split-off mass | $m_{split}$ | 300 | $\mathrm{kg\,m^{-2}}$ |
| Maximum mass of a box | $m_{max}$ | 500 | $\mathrm{kg\,m^{-2}}$ |
| Maximum number of snow layers | $n_{layers}$ | 15 | |
| Stefan-Boltzmann constant | $\sigma$ | $5.670373\times10^{-8}$ | $\mathrm{W\,m^{-2}\,K^{-4}}$ |
| Universal gas constant | $R$ | 8.314 | $\mathrm{J\,K^{-1}\,mol^{-1}}$ |
| Maximum liquid water content | $\zeta_{max}$ | [0.01, 0.05, 0.1] | 1 |
| Integration time step | $dt$ | 86,400 | s |

## 2.2 Firnification

As the snow accumulates it becomes denser due to the growing overburden pressure. We follow Schwander et al. (1997) for the firnification of snow. Because the densification processes differ for densities above and below 550 $\mathrm{kg\,m^{-3}}$, two different models are used. The equations used here were calibrated for dry snow, nevertheless, we apply them to wet snow too.

5   Below 550 $\mathrm{kg\,m^{-3}}$, the rate of snow densification is proportional to the accumulation rate $A$ in units $\mathrm{kg\,m^{-2}\,s^{-1}}$ and a temperature dependent variable $k_0$ (Herron and Langway, 1980):

$$\frac{\partial \rho_s}{\partial t} = k_0 A (\rho_i - \rho_s), \tag{1}$$

where

$$k_0 = 0.011 \ \mathrm{m^2\,kg^{-1}} \exp\left(\frac{-10,160 \ \mathrm{J\,mol^{-1}}}{R \cdot T_s}\right). \tag{2}$$



Modifying the original parameterization for dry snow, here both snowfall and rain count toward the accumulation.

Densification beyond 550 kg m$^{-3}$ follows Barnola et al. (1991). This model is more physical because it takes into account the overburden pressure and not only the implied change in pressure due to accumulation. Here, the density change is given by

$$\frac{\partial \rho_s}{\partial t} = k_1 \rho_s f \cdot \Delta p^3, \tag{3}$$

where

$$k_1 = 25,400 \text{ MPa}^{-3} \text{ s}^{-1} \exp\left(\frac{-6 \cdot 10^4 \text{ J mol}^{-1}}{R \cdot T_s}\right). \tag{4}$$

As $k_0$ above, $k_1$ is an empirical, temperature dependent variable. $\Delta p$ is the pressure difference between the overburden pressure and the inside gas pressure. The later one only occurs after bubble close off and thus is not relevant here. $f$ is a dimensionless factor, which for densities $\rho_s$ between 550 kg m$^{-3}$ and 800 kg m$^{-3}$ $f$ is given by

$$\log_{10}(f) = \beta\left(\frac{\rho_s}{\rho_i}\right)^3 + \gamma\left(\frac{\rho_s}{\rho_i}\right)^2 + \delta\left(\frac{\rho_s}{\rho_i}\right) + \epsilon, \tag{5}$$

with the empirical dimensionless parameters $\beta = -29.166$, $\gamma = 84.422$, $\delta = -87.425$, and $\epsilon = 30.673$. Finally, although this case rarely applies in the current configuration with 15 vertical layers, for densities $\rho_s$ above 800 kg m$^{-3}$ $f$ is equal to

$$f = \frac{3}{16}\left(1 - \left(\frac{\rho_s}{\rho_i}\right)\right) / \left(1 - \left(1 - \left(\frac{\rho_s}{\rho_i}\right)\right)^{1/3}\right)^3. \tag{6}$$

### 2.3 Energy balance

#### 2.3.1 Surface energy flux

The surface energy balance takes into account four different ways of exchanging energy with the atmosphere: Thermal longwave radiation $Q_{lw}$, downwelling shortwave radiation $Q_{sw}$, sensible heat exchange with atmosphere $Q_{sh}$ and heat transport by precipitation $Q_p$. The equation for the change in temperature of the surface box due to surface fluxes then reads

$$c_i m_{s,1} \left.\frac{\partial T_s}{\partial t}\right|_{\text{surface}} = Q_{sw} + Q_{lw} + Q_p + Q_{sh}, \tag{7}$$

with the mass per area of the surface box $m_{s,1}$ and the heat capacity of ice $c_i$.

**Shortwave radiation:** Shortwave radiation depends only on the snow albedo ($\alpha_{\text{snow}}$) and the incoming shortwave radiation at the bottom of the atmosphere ($S_{\text{BOA}}$) which is an input parameter provided by the climate forcing:

$$Q_{sw} = (1 - \alpha_{\text{snow}}) S_{\text{BOA}}. \tag{8}$$

The albedo of snow changes considerably when melting occurs and thus has an impact on how much energy of the incoming solar radiation at the bottom of the atmosphere, $S_{\text{BOA}}$, is absorbed by the surface of the snow cover. This is taken into account by using two different values for $\alpha_{\text{snow}}$, $\alpha_{\text{dry}}$ for dry snow with temperatures below the freezing point and $\alpha_{\text{wet}}$ for wet snow with a temperature $T_s = 0°\text{C}$. A wide range of values is plausible depending on the surface conditions, 0.7-0.98 for dry snow,

0.46-0.7 for wet snow, and for ice 0.3-0.46 (Paterson, 1999). See Table 2 for values tested.

**Longwave radiation:** The net longwave radiation at the snow surface is the difference between the downward directed part from the atmosphere and the upward radiation emitted by the snow:

$$Q_{\text{lw}} = \sigma(\epsilon_{\text{air}} T_{\text{air}}^4 - \epsilon_{\text{snow}} T_{\text{s}}^4), \tag{9}$$

where $\sigma$ is the Stefan-Boltzmann constant, $\epsilon_{\text{air}}$ is the emissivity of air and $\epsilon_{\text{snow}}$ is the emissivity of snow. $T_{\text{air}}$ is the air

temperature as provided as a boundary condition from the climate forcing. The emissivity of snow is $\epsilon_{\text{snow}} = 0.98$. Clouds and air humidity generally have a large impact on the net longwave radiation balance due to their high emissivity in the infrared spectrum but also the air temperature influences the emissivity of air. This is reflected in the great spread for measured values for $\epsilon_{\text{air}}$. Busetto et al. (2013) suggest $\epsilon_{\text{air}}$ values between 0.4 and 0.7 under the very dry conditions on the Antarctic Ice Sheet, measurements in Greenland suggest that $\epsilon_{\text{air}}$ takes values between 0.7 and 0.9 (Greuell, 1992; Paterson, 1999). In general,

not only the surface air temperature is relevant for this energy flux, but also air temperatures higher up in the atmosphere. We acknowledge that the simplification to one global value for $\epsilon_{\text{air}}$ likely an oversimplification. However, the alternatives are using either the longwave downward radiation or the three-dimensional temperature and humidity fields and cloudiness as climate boundary conditions, which are not readily available from coarse resolution climate model that BESSI is designed to be coupled with.

**Sensible heat flux:** The bulk coefficient for the sensible heat flux $D_{\text{sh}}$ is (Braithwaite, 2009):

$$D_{\text{sh}} = 1.29 \times 10^{-2} \, \text{K}^{-1} \cdot A \, p \, u, \tag{10}$$

where $u$ is the wind speed, $p$ the average atmospheric pressure, and $A$ and empirical, dimensionless transfer coefficient. $A$ takes values between 1.4 and $3.6 \times 10^{-3}$. Assuming an air pressure of $p$ of $10^5$ Pa and a wind speed of $5 \, \text{m s}^{-1}$ $D_{\text{sh}}$ takes values between 9 and 23 W m$^{-2}$ K$^{-1}$. The sensible heat flux is given by:

$$Q_{\text{sh}} = D_{\text{sh}}(T_{\text{air}} - T_{\text{s}}). \tag{11}$$

**Heat transport with precipitation:** Snow and rain falling onto the snow generally do not have the same temperature as the firn and thus represent a heat sink or source.

$$\begin{aligned} Q_{\text{p,rain}} &= P \cdot \rho_{\text{w}} c_w (T_{\text{air}} - 273 \, \text{K}), \tag{12} \\ Q_{\text{p,snow}} &= P \cdot \rho_{\text{w}} c_i (T_{\text{air}} - T_{\text{s}}), \tag{13} \end{aligned}$$



where $P$ is the precipitation and $\rho_{\mathrm{w}}$ the density of water. $c_{\mathrm{w}}$ and $c_{\mathrm{i}}$ are the heat capacities of water and ice respectively. This sensible heat of the precipitation influences only the surface box of the snow column. The latent heat due to freezing and melting may affect deeper layers as well and is therefore treated separately and described below.

### 2.3.2 Diffusion of heat and latent heat

5 Below the surface, heat is transported by diffusion and as latent heat in liquid water. Temperature differences between the vertical layers cause diffusion of heat:

$$c_{\mathrm{i}}\rho_{\mathrm{s}}\frac{\partial T_s}{\partial t} \quad = \quad \frac{\partial}{\partial z}(K(\rho_{\mathrm{s}})\frac{\partial T_s}{\partial z}), \tag{14}$$

where $K(\rho_{\mathrm{s}})$ is the thermal conductivity of snow as a function of the density of snow (Yen, 1981):

$$K(\rho_{\mathrm{s}}) = K_i(\frac{\rho_{\mathrm{s}}}{\rho_{\mathrm{w}}})^{1.88}. \tag{15}$$

10 However, this approximation does not account for the effects of liquid water nor the influence of the snow structure.

The heat released by the melting of snow or freezing of water is:

$$Q_{\mathrm{lh}} \quad = \quad -L_{\mathrm{lh}} \cdot m_{\mathrm{s}}, \tag{16}$$

$$Q_{\mathrm{lh}} \quad = \quad L_{\mathrm{lh}} \cdot m_{\mathrm{w}}, \tag{17}$$

where $L_{\mathrm{lh}}$ is the specific heat of fusion. While melting may only occur at the surface, freezing of liquid water is allowed to 15 take place everywhere in the snow column.

## 2.4 Mass balance

### 2.4.1 Surface accumulation and ablation

**Accumulation:** Depending on the air temperature at each time step either snow or rain falls on the top grid cell. Generally all precipitation that falls below the freezing point of water is considered to be snow with a temperature equal to that of the air. 20 Snowfall is directly added to the mass of the first grid box as

$$\frac{\partial m_{s,1}}{\partial t} = P \cdot \rho_{\mathrm{w}}. \tag{18}$$

Precipitation at time steps with an air temperature above 0°C is treated as rain with air temperature and is added to the water content of the surface box:

$$\frac{\partial m_{w,1}}{\partial t} = P \cdot \rho_{\mathrm{w}}. \tag{19}$$





The temperature difference between rain and the freezing point causes an energy flux into the snow cover as detailed in section 2.3.1. All water in the pore volume of the snow has an assumed temperature of 0°C. Sublimation and windborne redistribution of snow are not included. While they are usually negligible in regions of relatively high accumulation, sublimation in central Greenland is estimated to be 12-23% of the precipitation under present day conditions (Box and Steffen, 2001).

**Ablation:** If the net incoming energy at the surface warms the surface layer to temperatures above the melting point, the temperature is kept at 0°C and the excess heat is used to melt snow (eq. 16). Should the top layer melt entirely, the remaining energy is used to heat the next layer to 0°C and then melt it. This process is repeated until the entire melt energy is consumed. The resulting melt water is added to the liquid water mass $m_w$ of the corresponding layers. In the case that the entire snow column melts, the remaining energy is used to melt ice. Bare ice is also assumed to be at 0°C. Water from melted ice is treated

as runoff and removed from the model immediately.

### 2.4.2    Percolation and refreezing of water

**Percolation:** Water may fill the pore volume in the snow up to a certain maximum fraction $\zeta_{max}$ (Greuell, 1992). Water in excess of this maximum percolates downwards to the next box where the same fraction applies. Water that leaves the lowest active box is deemed runoff and removed from the model. The fractional liquid water content of a grid cell is calculated as:

$$\zeta = \frac{m_w}{m_s} \frac{1}{\rho_w \cdot \left( \frac{1}{\rho_s} - \frac{1}{\rho_i} \right)}. \tag{20}$$

For snow densities close to that of ice $\rho_i$, numerical problems arise due to division by (almost) zero. Thus, all water percolates downwards if the grid cells density satisfies $\rho_s > \rho_i$ - 10 kg m$^{-3}$. This happens in areas with a lot of rain and melt water that freezes again during the winter. Measurements at the ETH Camp in Greenland suggest a value of 0.1 for $\zeta_{max}$, but also lower values are used (Greuell, 1992).

**Refreezing:** If water percolates into colder layers or when temperatures decrease, it may freeze again leading to internal accumulation. Since the time steps are relatively long and the pores in the firn are rather small, we assume that the liquid water is in thermodynamic equilibrium with the surrounding snow. If the snow is not cold enough, it is possible that water freezes only partly. In this case the snow is heated to 0°C and the energy $(273 \text{ K} - T_s) \cdot c_i \cdot m_s$ is used to freeze the corresponding amount of water (eq. 17). If all water freezes and the snow has now the following temperature

$$T'_s = \frac{m_w \cdot L_{lh}/c_i + 273 \text{ K} \cdot m_w + T_s \cdot m_s}{m_w + m_s}. \tag{21}$$

In both cases, snow density $\rho_s$, mass $m_s$, and the mass of water $m_w$ are adjusted accordingly, conserving the volume of the grid box.



### 2.4.3 Surface mass balance for the ice model

If the total mass of the entire snow column exceeds $m_{\text{column}} = 1.5 \cdot m_{\text{split}} \cdot n_{\text{layers}}$ at the end of the year, the surplus mass is removed from the bottom and transferred to the ice model. The factor of $1.5$ is applied so that rarely snow is removed from the second lowest layer and only in snow columns where significant amounts of refreezing occurs. In perennially cold areas the lowest grid box grows very thick. Transferring the surplus accumulation from each snow column results in an even field of accumulation being passed on to the ice model as compared to passing full boxes whenever they reach a certain threshold. This avoids numerical instabilities in the ice dynamics code.

### 2.5 Climate input data

BESSI only requires precipitation, shortwave radiation at the bottom of the atmosphere, and surface air temperature as boundary conditions. With the exception of section 5, these fields are taken from the ERA-Interim climate reanalysis (Dee et al., 2011). The forcing consist of daily averages for 38 years, from 1979 to 2016. Since the climate forcing data are not referenced to the same topography as the one in BESSI, the temperature field is corrected with a constant lapse rate of $0.0065\text{K m}^{-1}$. Shortwave radiation and precipitation are not corrected, although changes in elevation modify the optical depth of the atmosphere and the transport of moisture. The temporal resolution of the input data is 365 time steps per year. The model has also been tested with a longer time step of 96 per year, which corresponds to the time step of the Bern3D coarse resolution climate model (Ritz et al., 2011). The longer time step produces good results but they have not been tested thoroughly and are therefore not included here.

### 2.6 Numerical implementation

BESSI is implemented in FORTRAN90 as a subroutine of the ice sheet model IceBern2D (Neff et al., 2016). The decision tree and calculation flow at each time step is shown in Fig. 4. The model iterates through all grid cells on land with a time step of one day. At every time step, calculations are executed for every active snow layer if not specified otherwise.

The first step during every iteration is accumulation at the surface. Depending on the air temperature either snow or rain is accumulated and added to the corresponding mass variable of the surface box (Sec. 2.4.1). The accompanying heat transport is stored as an energy flux which will be used later during the time step along with all other energy fluxes. At grid boxes where there is no snow present after this step, the potential melt of ice is calculated with the energy that enters a hypothetical ice layer during that time step. In these grid boxes, the time step ends here.

Where there is snow present, the layers are regridded if necessary (Sec. 2.1), and the densification is calculated (Sec. 2.2). In order to save calculation time, only columns with three and more layers of snow are densified. Most cells that only carry seasonal snow do not need more than two active snow boxes, and densification is negligible where the snow disappears completely every year.

The surface energy balance and diffusion within the snow (eq. 7 and 14) are solved in a single step using an implicit leap frog scheme. However, this may heat the snow to temperatures above $0°C$. When this happens, the result is discarded and the calculation is repeated without surface heat flux but with a fixed surface snow temperature of $0°C$. The energy necessary to heat




the surface box to the melting point is subtracted from the available melting energy, to ensure energy conservation. Melting only occurs at the surface and the corresponding amount is added to water mass of the uppermost box. However, during one time step more than one snow layer may melt and the vertical grid is adjusted accordingly.

Now, the percolation of the liquid water in the snow layers is calculated. This includes both melt water and rain. The routine
starts at the top box and works down all active boxes. Water only travels downwards and at the end of this step no grid box contains more water than the fraction $\zeta_{max}$ of the pore volume (Sec. 2.4.2). Some of the percolated water may refreeze in the snow. This process can accelerate the densification process of the firn and it modifies the heat profile.

Finally, excess mass is removed from the bottom of the snow column at the end of each model year and combined with the accumulated potential melt from the beginning of all time steps to calculate the mass balance for the ice model. The integrated
balance is transferred to the ice model. From here on the procedure is repeated with the input parameters from the next time step.

## 3   Model assessment and calibration

### 3.1   Conservation of mass and heat

In order to verify energy and mass conservation in BESSI, all mass and energy fluxes into and out of the snow cover are summed
up separately (diagnosed) and compared with the effective changes in snow mass and snow energy content simulated (observed) by the model. This is done in a simulation that is forced with climatological daily averages created from the 1979–2016 ERA-interim climatology. After 5000 model years, the snow model in in equilibrium. There are $15 \cdot 313 \cdot 313 \approx 1.5 \times 10^6$ grid cells on the model domain each being calculated at FORTRAN double precision of $10^{-16}$. Thus the maximum relative error of total mass and energy fluxes on the entire domain with 64 bit machine precision should not be larger than $1.5 \times 10^6 \cdot 10^{-16} \approx 10^{-10}$.
**Conservation of mass:** Mass is added to the snow cover as snow and rain. The model loses mass by percolation of water out of the lowest active box or by passing mass to the ice model. The comparison of observed changes in the amount of snow and water in the model domain, and the sum of the components of mass fluxes shows that the relative error is largest during winter and spring when the snow mass is at its maximum, but never above $10^{-12}$ (Fig. 5). Over the 5000 year simulation time, the average unaccounted mass flux for the entire model domain is –2.09 kg a$^{-1}$. For liquid water, the relative error is less than
$10^{-13}$ with an average loss of –1.23 kg a$^{-1}$ for the entire domain. Thus, we conclude that the model conserves mass within computational accuracy.

**Conservation of energy:** The energy fluxes into and out of the snow model comprise surface energy fluxes, latent heat of snow and rain, and by snow and water that leaves the model at the lower boundary. Internal fluxes such as the thermal diffusion and the redistribution of latent heat by the liquid water must not change the energy content.
Indeed, the relative error between the observed changes in heat content and the sum of the fluxes at the upper and lower boundaries is $10^{-12}$ and thus also conserved to machine precision (Fig. 6). The average loss of heat is –167,917.7 J a$^{-1}$, equivalent to melting about 0.5 kg of ice.



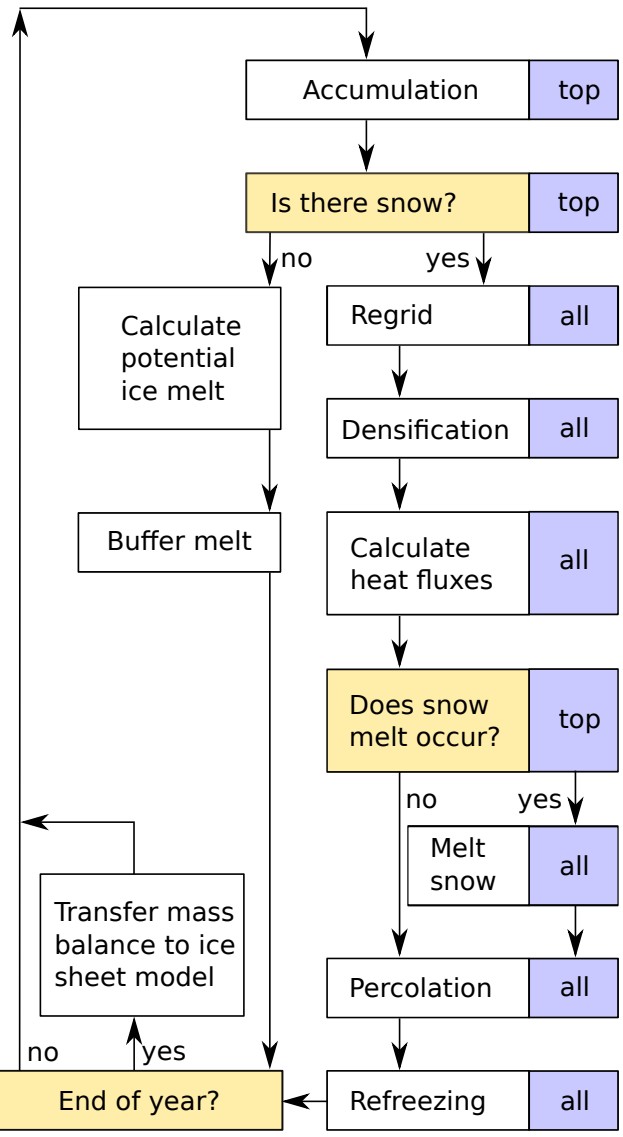

**Figure 4.** Decision tree at each time step. The blue boxes denote whether the process or question concerns only the top layer or all layers.

## 3.2 Model calibration

To reduce the parametric uncertainty of BESSI, a large ensemble of simulations is carried out based on the five most uncertain parameters. The calibration uses observations of the annual mean 10 m firn temperature from Greenland, annual cumulative Greenland mass loss data, and the monthly extent of the snow cover of the entire model domain. This selection is made to

5   assess both the energy and mass balance of the model, for both perennially and seasonally snow-covered regions.





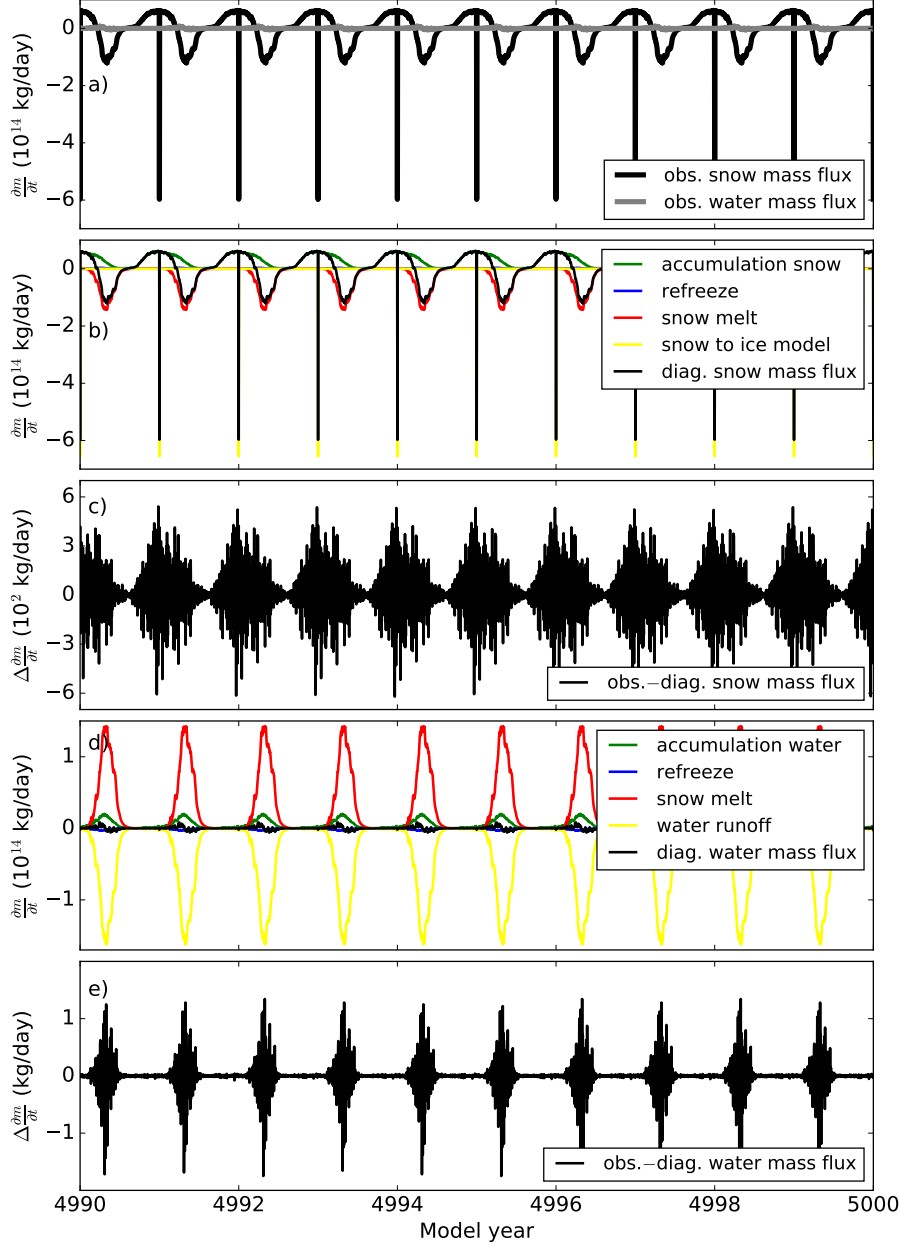

**Figure 5.** Total mass fluxes summed up over the entire model domain 5000 years after initialization. a) Observed changes of snow and water mass inside the snow model. b) Diagnosed fluxes in snow mass with all its contributing terms. c) Observed minus diagnosed snow mass fluxes. d) Diagnosed fluxes in water mass with all its contributing terms. e) Observed minus diagnosed water mass fluxes.

To calibrate the flux of heat we compare the model results with 23 firn temperature measurements from Greenland that were taken between 1996 and 2013 (Tab. 3). A similar approach was taken by Steger et al. (2017) with the measurements



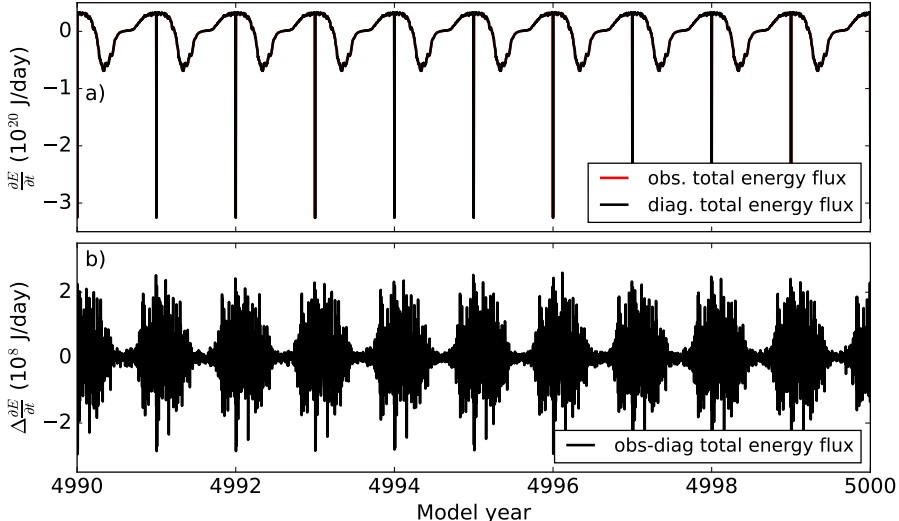

**Figure 6.** Total energy fluxes integrated over the entire model domain after 5000 model years. a) Observed and diagnosed total energy flux, the former being covered by the latter. (b) observed minus diagnosed total energy flux.

of Polashenski et al. (2014). Here we extend this data set with firn temperature measurements from Steffen and Box (2001), Schwander (2001), and Schwander et al. (2008) to achieve a broader spatial and temporal coverage of the GrIS. The 10 m firn temperature measurements of Steffen and Box (2001) are part of the Greenland Climate Network (GC-Net) and not maintained regularly, thus leading to poor control over the depth of the firn temperature sensors. Therefore, only a selection of measure-

ments is used that were taken less than three years after activation of the measurement site. Two additional measurements were included from the North Greenland Ice Core Project (NGRIP) and the North Greenland Eemian Ice Drilling (NEEM) sites (Schwander, 2001; Schwander et al., 2008). Although these were taken in summer, they are interpreted as representative of the annual average because the 10 m temperatures of GC-Net vary by less than 0.5°C over the annual cycle.

The seasonal snow cover is compared with satellite data starting in January 1979 until December 2016 (Northern Hemisphere

EASE-Grid 2.0 Weekly Snow Cover and Sea Ice Extent, Version 4; Brodzik and Armstrong (2013)). The weekly data are downsampled to a monthly resolution and interpolated onto the grid of BESSI. In the model, a grid cell is counted as covered with snow when the snow mass per area $m_s$ is more than $25 \, \mathrm{kg} \, \mathrm{m}^{-2}$, about 1 cm of fresh snow.

Changes in the total mass of the Greenland ice sheet are compared with measurements from the Gravity Recovery and Climate Experiment (GRACE; Watkins et al. (2015); Wiese et al. (2016)). Observed changes in surface mass balance are

calculated by subtracting the estimated increase in calving since 1996 (van den Broeke et al., 2016). We used cumulative annual averages for the years 2002 to 2018.

The five model parameters that are chosen to define the ensemble are the bulk coefficient for sensible heat flux $D_{\mathrm{sh}}$, the albedo of dry snow $\alpha_{\mathrm{dry}}$, the albedo of wet snow $\alpha_{\mathrm{wet}}$, the air emissivity $\epsilon_{\mathrm{air}}$ and the maximum water content ratio of the pore volume $\zeta_{\mathrm{max}}$. Each parameter is perturbed within its realistic range (Tab. 2) for a total of 756 simulations.



**Table 3.** Measurements of annual average firn temperatures at a depth of 10 m (see map in figure 1).

| Location | elevation [m a.s.l.] | year | $T_{10\ m}$ [°C] | source |
|---|---|---|---|---|
| Crawford P1 | 2022 | 1996 | −18.44 | Steffen and Box (2001) |
| Tunu-N | 2113 | 1996 | −28.87 | Steffen and Box (2001) |
| DYE-2 | 2165 | 1996 | −16.67 | Steffen and Box (2001) |
| Saddle | 2559 | 1999 | −20.32 | Steffen and Box (2001) |
| South Dome | 2922 | 1999 | −21.29 | Steffen and Box (2001) |
| Tunu-N | 2113 | 1999 | −28.63 | Steffen and Box (2001) |
| DYE-2 | 2165 | 1999 | −16.47 | Steffen and Box (2001) |
| NASA-E | 2631 | 1999 | −29.45 | Steffen and Box (2001) |
| Crawford P2 | 1990 | 1999 | −17.61 | Steffen and Box (2001) |
| NASA-SE | 2360 | 1999 | −19.34 | Steffen and Box (2001) |
| NGRIP | 2960 | 2001 | −31.19 | Schwander (2001) |
| NEEM | 2484 | 2008 | −28.10 | Schwander et al. (2008) |
| B 4-275 | 3071 | 2013 | −29.39 | Polashenski et al. (2014) |
| B 4-225 | 2996 | 2013 | −29.91 | Polashenski et al. (2014) |
| B 4-175 | 2949 | 2013 | −31.13 | Polashenski et al. (2014) |
| B 4-100 | 2860 | 2013 | −31.78 | Polashenski et al. (2014) |
| B 4-050 | 2781 | 2013 | −30.23 | Polashenski et al. (2014) |
| B 4-000 | 2664 | 2013 | −31.12 | Polashenski et al. (2014) |
| B 2-200 | 2540 | 2013 | −28.69 | Polashenski et al. (2014) |
| B 2-175 | 2445 | 2013 | −25.87 | Polashenski et al. (2014) |
| B 2-125 | 2198 | 2013 | −24.95 | Polashenski et al. (2014) |
| B 2-070 | 1971 | 2013 | −21.13 | Polashenski et al. (2014) |
| B 2-020 | 1905 | 2013 | −23.29 | Polashenski et al. (2014) |

All simulations are forced with daily average data from the ERA-Interim climate reanalysis for the period 1979–2016 (Sec. 2.5). For each parameter combination BESSI is initialized with snow by applying the climatic forcing of the years 1979–1998 forward and backward 6 times. The last 18 years are not used for the initialization because of their warming trend. The aim of the initialization sequence is to attain a firn that most closely resembles conditions at the beginning of the year 1979. After 5 these 240 years of initialization BESSI is fully equilibrated in the sense that all snow columns with a positive mass balance are filled with snow and forward mass to the ice sheet domain of the model. Subsequently, the years from 1979 to 2016 are simulated.



**Table 4.** Parameter combinations averaged over the ten simulations with lowest RMSE for monthly snow area $A$, Greenland borehole temperature $T$, and Greenland mass balance $m$, and the single simulation with the lowest RMSE for each metric (BEST$_X$).

|            | $\alpha_{\text{dry}}$ | $\alpha_{\text{wet}}$ | $D_{\text{sh}}$ (W m$^{-2}$ K$^{-1}$) | $\epsilon_{\text{air}}$ | $\zeta_{\text{max}}$ |
|------------|------|------|------|------|------|
| $T$        | 0.80 | 0.53 | 6    | 0.74 | 0.07 |
| $m$        | 0.81 | 0.57 | 12.5 | 0.74 | 0.06 |
| $A$        | 0.78 | 0.61 | 5    | 0.82 | 0.08 |
| BEST$_T$   | 0.80 | 0.5  | 5    | 0.75 | 0.1  |
| BEST$_m$   | 0.75 | 0.6  | 5    | 0.75 | 0.05 |
| BEST$_A$   | 0.75 | 0.7  | 5    | 0.85 | 0.1  |

The quality of all simulations is quantified by calculating the root-mean-square error (RMSE) of the modelled data:

$$\text{RMSE} = \sqrt{\frac{1}{i}\sum_{x=1}^{i}(X_{\text{model, i}} - X_{\text{obs, i}})^2}, \tag{22}$$

where $i$ is the number of observations, and $X_{\text{model, i}} - X_{\text{obs, i}}$ is the difference between simulated and observed data. The number of observations $i$ corresponds to the number of borehole sites (23), the years of mass loss measurements (16), and the

total number of grid points in the domain multiplied by the number of months in the observational record for the snow cover observations (43,498,236).

Table 4 lists the parameter values that lead to the best agreement with observations, i.e., the average values of the ten simulations with lowest RMSE. The ideal parameters for all three metrics are reasonably close with the exception of the bulk transfer coefficient for sensible heat $D_{\text{sh}}$. Additional detail can be found in the distributions of RMSE for single parameters

(Fig. 7). The atmospheric emissivity $\epsilon_{\text{air}}$ has a pronounced effect on the width and the median of the distributions, with a clear minimum within the range of tested values. This minimum is at lower values for the firn temperature than for the extent of the seasonal snow cover or the Greenland mass balance, probably because the firn temperatures have a regional bias to high altitude regions on Greenland where relatively dry and thin air reduce the downward flux of longwave radiation.

$D_{\text{sh}}$ does not have a clear optimum value within the range of tested values. This indicates that higher values are possible,

perhaps due to wind speeds that exceed the $5\,\text{m s}^{-1}$ that were postulated for the estimate of $D_{\text{sh}}$ above. The reduction in the spread for higher values of $D_{\text{sh}}$ suggests that other forms of heat exchange with the atmosphere become comparably less relevant.

## 4   Comparison with observations

In this section we analyze one simulation in more detail and compare it to some key observations. The simulation chosen is

the one with the lowest RMSE for the borehole temperature, BEST$_T$, unless explicitly specified otherwise (Tab. 4). Regions





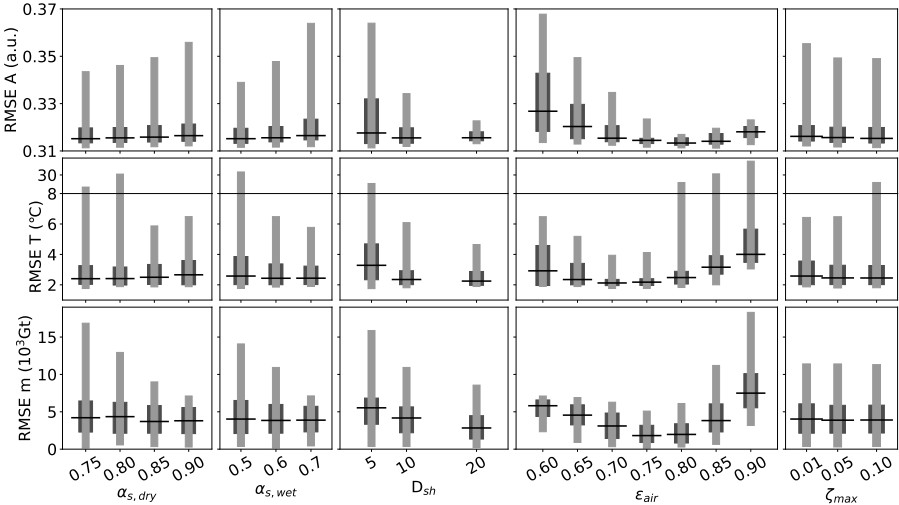

**Figure 7.** RMSE for monthly snow area A (top row), Greenland borehole temperature T (middle row, note the broken vertical scale), and Greenland mass balance m (bottom row), for five model parameters. Dark gray ranges contain 50% of the ensemble simulations, light gray 90%. The horizontal black line shows the median.

of net positive mass balance in BESSI compare well to glaciated areas, including ice caps in Canadian and Russian Arctic, southeast Iceland, southeast Alaska, Svalbard and the Pamir and Karakoram ranges (Fig. 1). Regions where the glaciation is below the grid size of the model are not reproduced, such as the European Alps, the Caucasus Mountains and the coastal parts of Scandinavia.

BESSI simulates the seasonal snow cover with adequate accuracy including intermittent snow events such as the one that covered central Europe in mid-December 2012 (Fig. 8). During the melting season, large parts of the snow contain liquid water. On Greenland, liquid water starts to appear on the margins in early summer (Fig. 8b) and gradually moves to higher elevations. At the beginning of the cold season, water that is close to the surface freezes again but above 2000 m elevation this process is slower due to the thermal insolation of the relatively thick firn which leads to two banded regions of liquid water

most notably in west Greenland (Fig. 8c). In some regions the water never freezes so that water is present even at the end of the cold season (Fig. 8a). This mechanism and the region in which it occurs are captured realistically by BESSI when compared to more sophisticated models and observations (Forster et al., 2014). However, BESSI probably underestimates the extent of the so-called perennial firn aquifer, because it only simulates the upper 15 m of the firn column.

      Although the model is designed for efficiency and simulations over multiple millennia, short-term events are represented

with reasonable accuracy as well. Most of the surface of the Greenland ice sheet experienced melting during the extreme melt season of 2012. BESSI cannot directly capture these melt events of relatively short duration, because its surface grid box is about 1 m thick and layers of melt are typically only a few centimeters thick. To accommodate this fact but nevertheless be able to qualitatively compare our simulation with observations, we define a melt day as the surface temperature in the model exceeding -5°C. Using this metric, BESSI agrees well with microwave radiometry observations (Fig. 9) (Mote, 2014).




The firnification process also compares well with the few available observations (Fig. 10). The version used here is limited to 15 vertical layers for efficiency, and so only represents the approximately linear increase in firn density. However, a version of BESSI with extended bottom also captures the curvature in the depth-density profile at around $550 \, \mathrm{kg \, m^{-3}}$ (not shown). The mismatch of the density at the surface is due to the density of the newly accumulated snow, which is fixed in the model.

The simulated surface mass balance of the Greenland ice sheet compares well with more complex models (Fig. 11 a) (van den Broeke et al., 2016). Here we show results from simulation $\mathrm{BEST}_m$. The total mass flux changed from around 0 Gt/yr before 1990 to -400 Gt/yr in 2015. This decrease is mostly due to an increase in surface melting, from -400 Gt/yr to -700 Gt/yr, while accumulation remained largely stable and refreezing increased slightly. The cumulative mass flux compares well with the calving-corrected estimate from GRACE (Fig. 11 b), as expected because this simulation was optimized toward this metric.

However, simulation $\mathrm{BEST}_T$ also shows a good agreement with GRACE in spite of being tuned to the Greenland borehole temperatures. Lastly, simulation $\mathrm{BEST}_A$ fails this test with a much too negative mass balance. The comparison of the model parameters (Tab. 4) shows that the main difference between the three simulations is the value of $\epsilon_{\mathrm{air}}$, which is 0.75 for the simulations optimized for Arctic climate of Greenland ($\mathrm{BEST}_m$, $\mathrm{BEST}_T$) and 0.85 for $\mathrm{BEST}_A$. The latter is sensitive to the extent of the seasonal snow cover, which mostly depends on the midlatitudes where the atmosphere contains more moisture

and clouds. The distribution of RMSE A clearly shows a minimum at higher values of $\epsilon_{\mathrm{air}}$ than for RMSE T or RMSE m (Fig. 7).

## 5   Nonlinearity of the surface mass balance

The numerical efficiency of BESSI allows for relatively fast calculations of large ensembles. Here we take advantage of this feature to quantify the nonlinearity of the surface mass balance with regard to anomalous temperatures.

It is generally accepted that the mass balance-temperature relationship has a negative curvature, meaning that positive temperature anomalies result in a stronger reduction of the mass balance than a corresponding temperature anomaly of the opposite sign would increase the mass balance (Fettweis et al., 2008; Mikkelsen et al., 2018). However, a higher air temperature also increases its capacity to hold and transport moisture, resulting in higher precipitation rates that may increase accumulation and thus counteract the more intense melting.

BESSI is initialized in the same way as before, where the forcing with the ERA interim climate reanalysis data from 1979 to 1998 is run six times forward and backward. After that, the model is run for the years 1979 to 1999, where a climate anomaly is added to the final year. This anomaly is defined by the annual average temperature difference over Greenland of a given year in a simulation of the Community Earth System Model version 1 (CESM1) with respect to the reference period 1950-1979 in the same simulation. The CESM1 simulation ran with full historic forcing from the year 850 to 2100 (Lehner et al., 2015), of

which we here use every fifth year between 1400 and 1995, and every year from 2000, to achieve a relatively even coverage of the temperature anomaly range. The simulations with BESSI may either use both the temperature and precipitation anomalies ($\Delta$T and $\Delta$P), or the temperature anomalies alone ($\Delta$T only), which results in a total of 442 simulations with a temperature





anomaly range from -5°to +7.5°. In addition, we also run 17 simulations with spatially uniform temperature anomalies from -8°to +8°. The shortwave radiation is unperturbed and all simulations in this ensemble use the $BEST_T$ parameter settings.

As expected, the surface mass balance becomes increasingly negative for higher temperatures in the absense of accompanying factors, creating a curve with negative curvature (Fig. 12, green). Although more complex in their spatial structure, the

same finding applies to temperature anomalies derived from CESM1 (red). Finally, if both temperature and the corresponding precipitation anomalies are applied, the picture is less clear as the points on the $\Delta T$–$\Delta SMB$ plane are more dispersed. The curvature remains negative but the SMB may increase for small perturbations (black).

Repeating the regression of order two for each surface grid point yields maps for the linear trend and curvature (Fig. 13). The simulations with a constant temperature offset are not included in this analysis. The linear trend of the SMB is negative

everywhere on Greenland in the case that BESSI is forced only with anomalous temperatures. If the corresponding anomalies in precipitation are also taken into account, regions on the southeastern and western upper margins show an increase in mass balance with higher temperatures. This is because warm temperatures in these regions are often caused by the influx of relatively moist oceanic air masses. Similarly, the curvature of the temperature-SMB curve is negative everywhere on the ice sheet for temperature anomalies whereas the combination of anomalous temperature and precipitation forcing results in positive

curvatures in the interior. Interestingly, the largest negative curvature coincides with the region of positive trend, indicating that relatively small temperature anomalies cause an increase in SMB, the trend inverses quickly as soon as a certain threshold is passed. A similar pattern is seen for the integrated mass balance of the entire ice sheet (Fig. 12, black), which has important consequences for the stability of the ice sheet in a warmer climate.

## 6   Discussion and Conclusions

We developed and calibrated the new surface energy and mass balance model BESSI that is optimized for long simulation times and for coupling with intermediate complexity climate models. The model domain includes all land areas on the Northern Hemisphere that were glaciated during the Pleistocene ice ages, at a lateral resolution of 40 km. The model has 15 vertical layers on a mass-following grid that covers the upper 15 m, approximately. With its three-dimensional domain, the physically accurate representation of surface energy fluxes, internal diffusion of heat, and the explicit tracking of liquid water and refreezing,

BESSI combines features of complex SMB models (Reijmer and Hock, 2008; Steger et al., 2017) with a relatively coarse vertical resolution and a long time step. While it cannot match the numerical speed of empirical or more heavily parameterized models (Reeh, 1991; Robinson et al., 2010), BESSI simulates up to 300 models years per hour on a modern personal computer, which fulfills our goal of developing a model that is capable of simulating multiple glacial cycles within a defensible amount of time. Although not included here, preliminary tests with an asynchronous time step promise to increase the speed by another

order of magnitude. For this, the computation in regions with only seasonal snow and without liquid water is carried out only once every fifty years.

The simulated seasonal snow cover agrees well with satellite observations. Ice caps are reproduced well in different climate zones ranging from dry Arctic conditions in northern Greenland and on Svalbard to maritime climate on Iceland and in





southeast Alaska, and to the semiarid continental climate of the Karakoram mountains. This demonstrates the robustness of the model to different boundary conditions and encourages future simulations with glacial climate. The surface energy and mass balances of the GrIS compare well to estimates from remote sensing.

The computational efficiency of the model enables large ensembles of simulations for calibration or to test the sensitivity to changes in the boundary conditions. As an example, we tested how the surface mass balance over Greenland depends on variations in surface air temperature, and the combined effect of when these temperature anomalies are accompanied by meteorologically coherent variations in precipitation. One key result is that such an increase in temperature does not in all regions lead to a decrease in net accumulation because the warmth comes from moist air masses that also bring snow. In addition to this positive linear trend, large parts of the interior of the GrIS have a positive curvature for the mass balance-temperature relationship. This means that contrary to previous findings that only considered variability in temperature regardless of theirmeteorological implications (Mikkelsen et al., 2018), forcing the model with a variable climate may lead to a higher net accumulation as compared to forcing with the climatological average.

The calibration revealed that the parameters for the atmospheric emissivity $\epsilon_{air}$ and the exchange of sensible heat $D_{sh}$ have a strong influence on the model's performance. This is not ideal because both parameters simplify spatially heterogeneous processes with a globally uniform single value. This result points to some of the necessary compromises to build a fast model: A better representation of the sensible heat flux requires information on wind speed, and a more realistic simulation of the the downward longwave radiation is not possible without detailed information on the cloud cover and atmospheric moisture content. However, following our design goal to develop a model for simulations over multiple glacial cycles, these forcing fields were deliberately excluded, because for periods as far back as the last ice age they are only available from simulations of general circulation climate models. The computational cost of these climate models makes long simulation times infeasible. For the same reason, the physical processes of sublimation and windborne redistribution of snow are not included in BESSI.

*Author contributions.* AB and MI developed the model code with advice from TS. AB and MI designed and performed the experiments, analyzed the output, and wrote the manuscript. TS provided valuable input to the discussion at all stages.

*Competing interests.* The authors declare that they have no conflict of interest.

*Acknowledgements.* We are grateful to Jakob Schwander for his help with the firnification model and for providing snow density data for the NGRIP and NEEM sites. AB is supported by a Starting Grant from the Bergen Research Foundation. TFS acknowledges support by the Swiss National Science Foundation. MI is supported by the Swiss National Science Foundation, project n° 200021-162444.





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





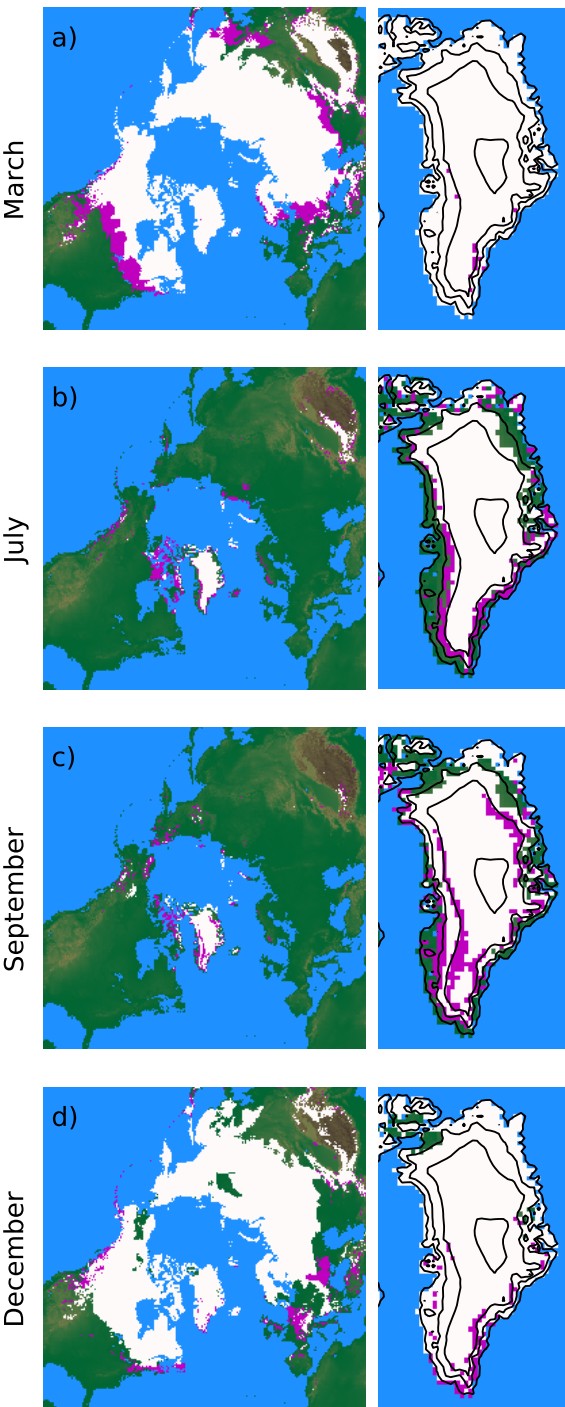

**Figure 8.** Seasonal snow extent for the year 2012, simulation $BEST_T$. White areas contained at least $25\ \mathrm{kg\,m^{-1}}$ snow at the middle of the respective month. Magenta highlights the presence of liquid water at any depth in the firn column. Contour levels show elevation with a spacing of 1000 m in the right column. Meltwater in the firn does not freeze immediately at the beginning of the winter season, which leads to the two bands of liquid water in west Greenland in September. Some regions never freeze all meltwater so that liquid water is present year round.



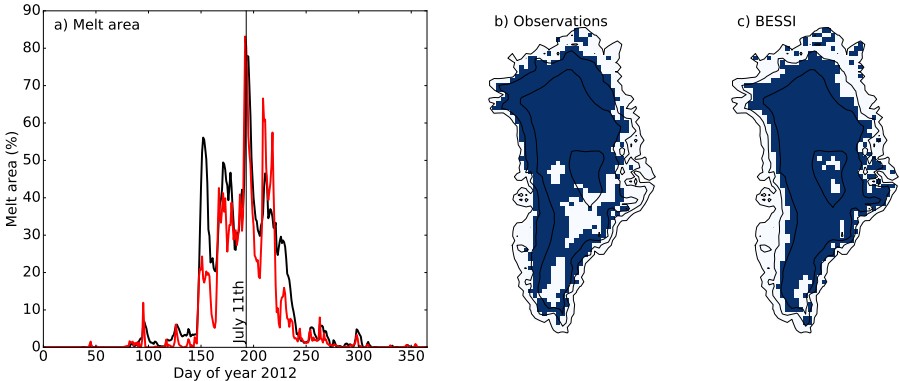

**Figure 9.** Melt extent of the year 2012, simulation $\text{BEST}_T$. a) Fractional melt area from observations (red) and BESSI (black), b) Observed melt area on July 11th, c) same as b) for BESSI.

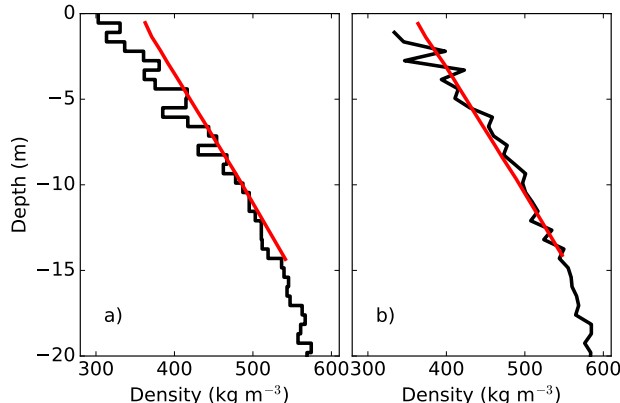

**Figure 10.** Modelled firn densities ($\text{BEST}_T$) at NGRIP (a) and NEEM (b) in red. Firn measurements of Schwander (2001) and Schwander et al. (2008) in black.



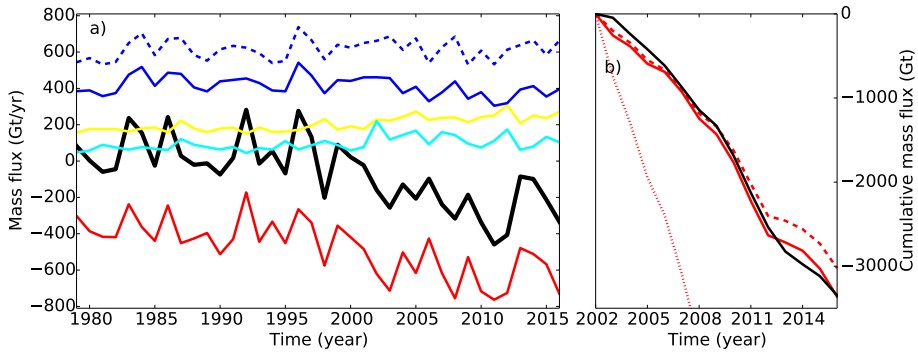

**Figure 11.** Surface mass balance as simulated by $BEST_m$. a) Total surface mass balance (black) and its components: accumulation (blue), total precipitation (blue dashed), melting (red), refreezing (light blue), and runoff (yellow). b) Cumulative annual mass balance for GRACE (black), $BEST_m$ (red), $BEST_T$ (red dashed), and $BEST_A$ (red dotted).

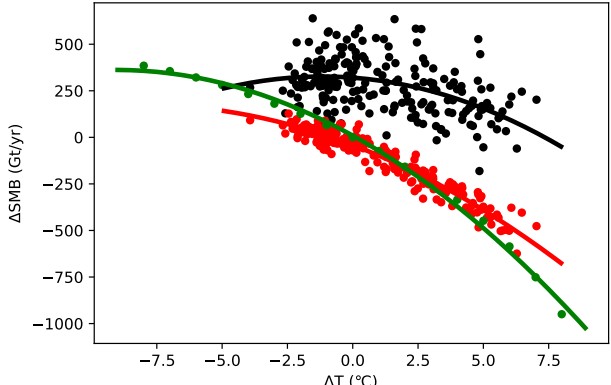

**Figure 12.** Change in surface mass balance as a function of anomalous annual average temperature over Greenland. Colors represent a spatially uniform temperature anomaly (green), a temperature pattern derived from a particular simulation year of a climate model (red), and the same temperature anomaly accompanied with the corresponding precipitation anomaly (black). Curves represent a least-squares regression of order two.





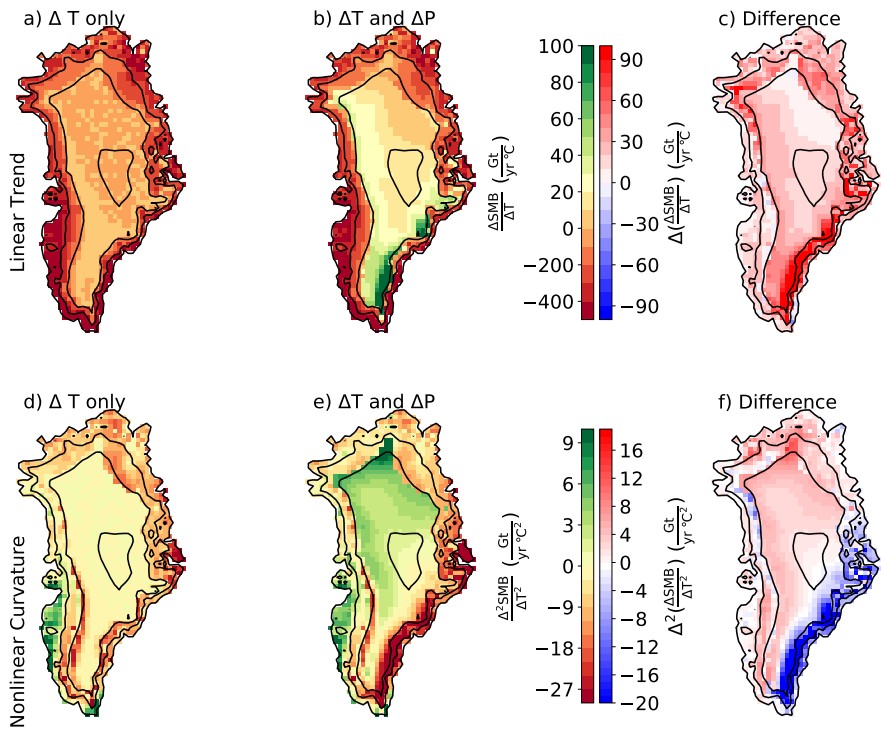

**Figure 13.** Spatial distribution of linear trend (a, b) and curvature (d, e) of the SMB anomalies as a function of perturbations in the annual mean temperature for simulations that were perturbed with the temperature anomalies alone (a, d), simulations that were perturbed by both anomalous temperature and corresponding precipitation (b, e), and the respective differences (c, f).