# Peer review of "An efficient surface energy-mass balance model for snow and ice"

_The Cryosphere, 2018_

## Referee Comment (RC1) · Krebs-Kanzow (Referee) · 10 Dec 2018

**1  General comments**

This manuscript presents the BErgen Snow SImulator (BESSI), a surface energy and mass balance model, which is designed to facilitate coupled ice sheet-climate simulations on multi-millennial time scales. In this context BESSI may serve as an interface between a coarse resolution and uncomprohensive forcing typically stemming from climate models of intermediate complexity and three-dimensional ice sheet models which require a detailed and accurate forcing consisting of mass and energy fluxes. Other common surface mass balance (SMB) models aiming at paleo-climate questions only consider a single snow layer (Krapp et al., 2017; Robinson et al., 2010) or

neglect processes in the snow pack and instead parameterize melting and refreezing empirically (Reeh, 1989). BESSI particularly sets itself apart from these schemes by representing snow and firn in 15 layers. Besides accumulation, the scheme uses insolation and near surface temperatures as a forcing and calculates the surface energy balance distinguishing the albedo of ice and dry and wet snow. Melting is deduced from the energy balance of the surface layer, while refreezing of liquid water is considered in all layers of the snow column. Furthermore, the heat diffusion equation and a firnification scheme yield temperature, snow mass, snow density and water content as prognostic variables in each model layer.

Born et al. provide a detailed model description, propose calibrations based on different data sets, and present a first application by investigating the sensitivity of Greenland's SMB to perturbed temperature and precipitation input.

The paper is generally well written, provides a good insight into snow pack modelling for a wider community, and the sophisticated snow pack representation is clearly a valuable contribution for the ice sheet modelling community. However, the paper also has some shortcomings which, in my view, would require major revisions.

**2  Major comments**

Abstract:

In its first half (lines 1-8) the abstract puts too much emphasis on motivation and background, while the second half is too short and unspecific (what is the calibration base, how was the model set-up evaluated, what is the time scale of the sensitivity experiments, what is the outcome of the sensitivity experiments...)

The benefits of a sophisticated representation of the snow pack could have been carved out better.

This aspect is missing in the introduction (e.g. are there biases and limitations in other schemes, which can be related to insufficient representation of processes in the snow column?). For the same reason, the results and discussion could particularly focus on snow covered regions and those processes which are important for coupled Earth system models.

The model does not resolve the diurnal freeze-melt cycles.
In my view, this is a major weakness in this scheme, which should be discussed. In Krebs-Kanzow et al. (2018) (Fig. 4) we demonstrate, that the length of the daily melt period influences surface melt rates. If the shape of the diurnal freeze-melt cycle is included in a SMB scheme, the same forcing (i.e. short wave energy uptake and air temperature) will result in different melt rates for different latitudes and seasons. Likewise nocturnal refreezing depends on the diurnal cycle. If the water holding capacity of the snow layer is sufficient to store the melt water produced during day-time, the net melt water production (or runoff) of the whole day will correspond to the melt rate predicted by a scheme, which uses only daily means. In any other case with distinct nocturnal refreezing, however, I would expect that such a daily scheme would underestimate the runoff, particularly over bare ice. I think it would be helpful to show the spatial distribution of the SMB of the ERA-Interim period, ideally in comparison to a regional model with sub-daily timestep, such as MAR or RACMO. Also the seasonal evolution could be of interest.

While the scheme is carefully calibrated, the evaluation is too short, in my view. As mentioned above, I think it would be interesting to assess the model's ability to reproduce spatial and seasonal patterns, maybe focussing on the accumulation area. Additionally, is it possible to specifically compare the regions outside of Greenland to observations in greater detail (e.g. onset and end of melt period, representation of large glaciers)?

**3  Specific comments**

Page 1 lines 13-14: Please provide a rough estimate of the amount of water stored in polar ice sheets. Also the reference in this sentence is wrong: it should be the locking of water which lowers the sea level, not the ice sheets.

Page 1 lines 19-21: Please specify: high-frequency variability is interannual here?

Page 2 lines 1-2: I assume that the acceleration of mass loss is related to positive feedbacks, so you might change the order of the first 2 sentences of this page and drop the "moreover".

Page 2 lines 3-32: I would propose a slightly changed structure:
Page 2 lines 11-15: This paragraph could move to the end of this part while
Page 2 lines 16-28: could be positioned earlier, maybe around line 5.

Page 2 lines 7-10: I think, primarily the problem is, that these models have a too low spatial resolution to resolve the narrow ablation zone. Most SMB schemes actually simplify physics (or even replace physical parameterizations by empirical functions) for the benefit of a better spatial resolution. This typical approach should be highlighted here, since BESSI does provide better physics in terms of the snow pack and uses physical meaningful atmospheric parameterizations (I would expect that even EMICs will include a similar degree of complexity in the atmosphere, though).

Page 2 line 33: This sentence is hard to understand and phase transitions as part of the energy balance should be mentioned. Maybe: The energy balance of the snow column is calculated by considering the energy fluxes through the surface and diffusive heat fluxes to deeper layers, the latent heat of melting in the surface layer and refreezing in all layers.

Page 3 line 7: What is the reason to choose 15 layers?

Page 3 line 9: To me it is not clear, how this follows from the previous sentences; maybe it is better without "thus".

Page 8 line 15: Please replace surface temperature by near surface temperature

Page 12 line 15: The formulation "observed" could be misleading- maybe use something like "simulated" or "effective"

Fig. 4: Does the scheme only transfer the mass balance to the ice model or also heat flux/temperature?

Fig. 5 and 6: The model seems to be conserving mass and energy almost perfectly and I wonder if these figures could be reduced to fewer seasonal cycles, or even be replaced by some statistics, while the figures could be moved to the supplement.

Page 13 Model calibration: The choice of calibration data sets should be motivated. I guess that the calibration is deliberately limited to data which are direct and relatively precise measurements (with the exception of the surface mass balance time series deduced from GRACE). However, the ablation zone is not well represented in the calibration and consequently, an evaluation of the spatial pattern of the SMB is important (see major comments).

Page 17 lines 1-6 and Table 4: Here, the clarity could be improved. Maybe the parameter combination over the ten simulations with lowest RMSE could appear as $TOP10_x$ and together with $BEST_x$, could be introduced in the text before line 4.

Page 17 lines 14-17: I don't understand, why higher wind speeds might reconcile relatively low optimal parameters of $D_{SH}$. Also, at least over melting ice, wind speeds are rather reduced. And finally, the last sentence of this paragraph is not very clear to me.

Page 18 line 17-19: I don't find this analysis very convincing. I assume that surface temperatures are closely related to air temperature and even a PDD scheme would predict melt, if forced with daily temperatures $> -5^oC$.

Fig. 11a: What exactly is runoff? It does not seem to be runoff=rain+melting-refreezing.
Generally, I don't seem to interpret Fig. 11 correctly. I don't see a good agreement with van den Broeke et al. (2016), who estimate SMB $\approx$ 300-400 Gt, accumulation is $\approx$ 600 Gt and refreezing is $\approx$ 200 GT.

Page 20 line 2: Indeed, considering short wave radiation anomalies might be interesting. Is it possible to discuss this option?

**References**

Krapp, M., Robinson, A., and Ganopolski, A.: SEMIC: an efficient surface energy and mass balance model applied to the Greenland ice sheet, Cryosphere, 11, 1519–1535, https://doi.org/{10.5194/tc-11-1519-2017}, 2017.
Krebs-Kanzow, U., Gierz, P., and Lohmann, G.: Brief communication: An Ice surface melt scheme including the diurnal cycle of solar radiation, The Cryosphere Discussions, 2018, 1–11, https://doi.org/10.5194/tc-2018-130, https://www.the-cryosphere-discuss.net/tc-2018-130/, 2018.

[Figure]

Reeh, N.: Parameterization of melt rate and surface temperature on the Greenland ice sheet, Polarforschung, 59, 113–128, 1989.

Robinson, A., Calov, R., and Ganopolski, A.: An efficient regional energy-moisture balance model for simulation of the Greenland Ice Sheet response to climate change, The Cryosphere, 4, 129–144, https://doi.org/{https://doi.org/10.5194/tc-4-129-2010}, 2010.

van den Broeke, M. R., Enderlin, E. M., Howat, I. M., Kuipers Munneke, P., Noël, B. P. Y., van de Berg, W. J., van Meijgaard, E., and Wouters, B.: On the recent contribution of the Greenland ice sheet to sea level change, The Cryosphere, 10, 1933–1946, https://doi.org/10.5194/tc-10-1933-2016, https://www.the-cryosphere.net/10/1933/2016/, 2016.

---

## Referee Comment (RC2) · Anonymous Referee #2 · 3 Jan 2019

First of all, I want to apologize to the authors and the editor for my late reply but some unforeseen circumstances have made a quicker reply impossible.

This paper by Born and colleagues describes a multi-layer layer snowpack and energy balance models which is suitable for large icesheet simulations on multi-millennial time scales. The snowpack model (BESSI) is part of the IceBern2D model and it has been tested with climatological data (ERA-Interim) in terms of its energy and mass conservation. The only shortcoming of this fast and efficient snowpack model is that it strongly depends on the value of atmospheric emissivity and the sensible heat exchange coefficient. The paper itself is well structured and has been written in a clear way.

This paper is a valuable contribution to the Cryosphere Community specifically for

those interested in computationally efficient global snowpack models needed for interactive glacial–interglacial climate/icesheet model simulations of the past or large ensembles for future scenarios.

I would recommend this paper for publication in TC after the addressing the following remarks (my major point being the model/data availability, see below).

**Major Comments**

- My one major criticism of this paper is that the authors do not state if and how the model/data will be made available (this affects how I scored the significance of this paper). The journal's data policy requires the following: "If the data are not publicly accessible, a detailed explanation of why this is the case is required". The authors mention that BESSI is part of IceBern2D but because IceBern2D does not seem to be publicly available (as far as I can tell), I suspect that neither will be BESSI. If this is really the case, than, in my point of view, this would be a bad decision in the light of scientifc transparency and reproducibility. The Cryosphere Community would defineltly benefit from making every model code and associated data publicly available (you can't simply reproduce such a complex model from scratch). As a best practise example, see Krapp et al. (2017). However, this is a decision to be made by the editor and/or in general by the editoral board of TC and not by me as a reviewer so I leave the final decision here to the editor. (I am sorry if this sounds a bit harsh but I strongly feel that we as a community need to be more transparent with what we do and show and I sense that Open Access is just one part of Open Science; for example: https://www.practicereproducibleresearch.org).

- I find the title misleading. The authors do not show any glacial cycle simulation results.

- In the abstract you write "...even a marginal bias will develop into an erroneous solution over the long integration time and when amplified by strong positive feedback mechanisms": I am not sure that this is shown in the paper so please rephrase.

- In the introduction, what is the benefit of having mutiple layers compared to a single layer? Is it possible to run BESSI in "single layer mode"; this would then allow a direct comparison of the effect of a single vs. multi-layer snowpack model.

- Sect 2.1: As BESSI is integrated onto IceBern2D what is the reason that you didn't use the fully coupled version of IceBern2D/BESSI?

- What about the contribution due to latent heat exchange in Eq (7)?

- The parameterizations of the different terms in the surface energy balance (Sec 2.3.1) indicate a few uncertainties: snow albedo ($Q_{SW}$), atmospheric emissivity ($Q_{LW}$), and wind speed and air pressure ($Q_{SH}$)

  – What are the expected uncertainty ranges for each of those terms (as presented for $Q_{SH}$)

  – E.g., atmospheric emissivity varies with cloud cover, snow albedo varies with liquid water content or dust particles, wind speed varies non-uniformly across an ice sheet, and air pressure changes with with ice sheet height

- What are the vertical jumps in Fig. 5 a) and b) and 6 a)?

- p.19, l.1: In principle, BESSI could also be evaluated against snow temperature profile data from mountain glaciers, e.g., Gilbert et al. (2016), their Fig. 6 a–d and Fig. 9 (firnification data)

- I don't see the added value of Sect 5; it is rather confusing for the reader to start again with another model setup with a different climate model; I can't see why

this section is important at all and thus this section fells short compared to the rest of the paper

**Minor Comments**

- accumulation rate $A$ and pressure change $\Delta p$ should be added to Table 1

- replace 273K with $T_0 = 273.15\,K$ throughout the manuscript

**References**

- Krapp, M., Robinson, A., and Ganopolski, A.: SEMIC: an efficient surface energy and mass balance model applied to the Greenland ice sheet, The Cryosphere, 11, 1519-1535, https://doi.org/10.5194/tc-11-1519-2017, 2017.

- Gilbert, A., Vincent, C., Six, D., Wagnon, P., Piard, L., and Ginot, P.: Modeling near-surface firn temperature in a cold accumulation zone (Col du Dôme, French Alps): from a physical to a semi-parameterized approach, The Cryosphere, 8, 689-703, https://doi.org/10.5194/tc-8-689-2014, 2014.

---

## Author Comment (AC1) · 31 Jan 2019

We are grateful to both referees for their positive and constructive comments. It is our intent to address all of them in the revised manuscript. Here, we will reply to the major points that were raised in the discussion:

We agree that processes on time-scales shorter than one day are potentially important for the mass balance in certain regions and we will revise the manuscript to discuss how their omission may impact our results. Referee Krebs-Kanzow requested that we compare BESSI with spatial patterns from more sophisticated models. We refer to the recently published study by Plach et al. (2018), and Plach et al. (2019, in revision) where we did such a comparison with MAR for four different time slices ranging from

relatively cool to notably warmer climates. The main difference is that BESSI estimates much less refreezing than MAR, which can at least partially be explained by the lack of a diurnal cycle in our model. We will reference the Plach et al. papers in the revised manuscript. We will also weigh the benefits of a parameterization of diurnal melt-refreeze cycles in a future version of BESSI (Krapp et al. 2017; Krebs-Kanzow et al. 2018), and include these considerations in the discussion.

Regarding the comparison with observations from large glaciers outside Greenland, a suggestion made by both reviewers, we feel that the number of neglected processes becomes too large (slope angle and orientation, snow drift, micrometeorology, etc.) and the lateral resolution too large (40km) for this comparison to provide meaningful insight.

The request to make the model code publicly available is very reasonable and we will follow it. Our preferred solution would be to include the source code as a supplement to the final paper.

In response to comments from both reviewers, we will revise title and abstract, and better motivate the use of a multi-layered model. The advantages include the explicit simulation of meltwater percolation, and that of vertical heat transport.

A. Plach, K.H. Nisancioglu, S. Le clec'h, A. Born, P. Langebroek, Chuncheng Guo, M. Imhof and T.F. Stocker (2018): Eemian Greenland SMB strongly sensitive to model choice, Climate of the Past 14, 1463-1485 DOI: 10.5194/cp-14-1463-2018

A. Plach, K.H. Nisancioglu, P.M. Langebroek and A. Born (2018): Eemian Greenland ice sheet simulated with a higher-order model shows strong sensitivity to SMB forcing, The Cryosphere Discussions (in revision) DOI: 10.5194/tc-2018-225

---

## Author Response (AR1)

Referee #1, Uta Krebs-Kanzow

**General comments**

This manuscript presents the BErgen Snow SImulator (BESSI), a surface energy and mass balance model, which is designed to facilitate coupled ice sheet-climate simulations on multi-millennial time scales. In this context BESSI may serve as an interface between a coarse resolution and uncomprohensive forcing typically stemming from climate models of intermediate complexity and three-dimensional ice sheet models which require a detailed and accurate forcing consisting of mass and energy fluxes. Other common surface mass balance (SMB) models aiming at paleo-climate guestions only consider a single snow laver (Krapp et al., 2017: Robinson et al., 2010) or neglect processes in the snow pack and instead parameterize melting and refreezing empirically (Reeh, 1989). BESSI particularly sets itself apart from these schemes by representing snow and firn in 15 layers. Besides accumulation, the scheme uses insolation and near surface temperatures as a forcing and calculates the surface energy balance distinguishing the albedo of ice and dry and wet snow. Melting is deduced from the energy balance of the surface layer, while refreezing of liquid water is considered in all layers of the snow column. Furthermore, the heat diffusion equation and a firnification scheme yield temperature, snow mass, snow density and water content as prognostic variables in each model layer.

Born et al. provide a detailed model description, propose calibrations based on different data sets, and present a first application by investigating the sensitivity of Greenland's SMB to perturbed temperature and precipitation input. The paper is generally well written, provides a good insight into snow pack modelling for a wider community, and the sophisticated snow pack representation is clearly a

valuable contribution for the ice sheet modelling community. However, the paper also has some shortcomings which, in my view, would require major revisions.

**A- We thank Dr. Krebs-Kanzow for her positive evaluation, and for specific and relevant criticism.**

Major comments

Abstract:

In its first half (lines 1-8) the abstract puts too much emphasis on motivation and background, while the second half is too short and unspecific (what is the calibration base, how was the model set-up evaluated, what is the time scale of the sensitivity experiments, what is the outcome of the sensitivity experiments...)

**A- We revised the abstract to accommodate these suggestions.**

The benefits of a sophisticated representation of the snow pack could have been carved out better. This aspect is missing in the introduction (e.g. are there biases and limitations in other schemes, which can be related to insufficient representation of processes in the snow column?). For the same reason, the results and discussion could particularly focus on snow covered regions and those processes which are important for coupled Earth system models.

A- The introduction has been re-organized and partly rewritten in reply to comments below. We now include a motivation for why a multi-layer model may capture important aspects better than its single-layer alternatives. Our model is not designed as a comprehensive land surface model and so the coupling with earth systems models is only relevant to the extent that it affects the net snow balance, i.e., the existence or not of perennially glaciated areas. We therefore do not discuss seasonally snow covered regions in this manuscript but only use the seasonal extent of the snow cover as an indicator for the overall performance during the calibration.

The model does not resolve the diurnal freeze-melt cycles. In my view, this is a major weakness in this scheme, which should be discussed. In Krebs-Kanzow et al. (2018) (Fig. 4) we demonstrate, that the length of the daily melt period influences surface melt rates. If the shape of the diurnal freeze-melt cycle is included in a SMB scheme, the same forcing (i.e. short wave energy uptake and air temperature) will result in different melt rates for different latitudes and seasons. Likewise nocturnal refreezing depends on the diurnal cycle. If the water holding capacity of the snow layer is sufficient to store the melt water produced during day-time, the net melt water production (or runoff) of the whole day will correspond to the melt rate predicted by a scheme, which uses only daily means. In any other case with distinct nocturnal refreezing, however, I would expect that such a daily scheme would underestimate the runoff, particularly over bare ice. I think it would be helpful to show the spatial distribution of the SMB of the ERA-Interim period, ideally in comparison to a regional model with sub-daily timestep, such as MAR or RACMO. Also the seasonal evolution could be of interest.

A- We agree that the omission of daily temperature variations may have significant impact on the results. Closer inspection of the Greenland mass balance shows that refreezing is strongly underestimated in comparison to RACMO and MAR and the latitudinal signature of the bias suggests that this could be related to the diurnal melt-refreeze cycle. We updated and corrected figure 11 and added references to our recently published study by Plach et al. (2018), which compares maps of mass balance from BESSI and MAR. We also updated both the results (4) and discussions (6) sections, as detailed in our reply to the specific comment on figure 11 below.

While the scheme is carefully calibrated, the evaluation is too short, in my view. As mentioned above, I think it would be interesting to assess the model's ability to reproduce spatial and seasonal patterns, maybe focussing on the accumulation area. Additionally, is it possible to specifically compare the regions outside of Greenland to observations in greater detail (e.g. onset and end of melt period, representation of large glaciers)?

A- We added a new figure (12) with maps for the annual net surface mass balance, melting and refreezing, and discuss the results together with the updated figure 11. Given the caveats discussed in the revised manuscript, we find good agreement with RACMO. We feel that further detail such as seasonal patterns or an extended discussion of the results are beyond the intended scope of this manuscript, both because the Greenland ice sheet was not the main objective here and because a future study that is designed to address these aspects, with an updated version of our model including several new physical processes and a higher resolution on the Greenland domain, is currently underway (Zolles et al.; https://meetingorganizer.copernicus.org/EGU2019/EGU2019-5350.pdf).

For similar reasons, we would prefer not to extend the discussion to glaciers. Not also that even large glaciers are not well captured at the lower limit of the 40km resolution.

**Specific comments**

Page 1 lines 13-14: Please provide a rough estimate of the amount of water stored in polar ice sheets. Also the reference in this sentence is wrong: it should be the locking of water which lowers the sea level, not the ice sheets.

A- We rephrased this statement and updated the reference and added an estimate on the amount of water stored in ice sheets today.

Page 1 lines 19-21: Please specify: high-frequency variability is interannual here?

A- Yes, the text has been rephrased accordingly.

Page 2 lines 1-2: I assume that the acceleration of mass loss is related to positive feedbacks, so you might change the order of the first 2 sentences of this page and drop the "moreover".

A- We separate the well-known ice-elevation feedback from processes that impact the dynamics of the atmosphere. The latter may constitute a negative feedback. This has been clarified in the text.

Page 2 lines 3-32: I would propose a slightly changed structure: Page 2 lines 11-15: This paragraph could move to the end of this part while Page 2 lines 16-28: could be positioned earlier, maybe around line 5.

**A- Done**

Page 2 lines 7-10: I think, primarily the problem is, that these models have a too low spatial resolution to resolve the narrow ablation zone. Most SMB schemes actually simplify physics (or even replace physical parameterizations by empirical functions) for the benefit of a better spatial resolution. This typical approach should be highlighted here, since BESSI does provide better physics in terms of the snow

pack and uses physical meaningful atmospheric parameterizations (I would expect that even EMICs will include a similar degree of complexity in the atmosphere, though).

**A- This sentence has been removed.**

Page 2 line 33: This sentence is hard to understand and phase transitions as part of the energy balance should be mentioned. Maybe: The energy balance of the snow column is calculated by considering the energy fluxes through the surface and diffusive heat fluxes to deeper layers, the latent heat of melting in the surface layer and refreezing in all layers.

A- This part of the introduction has been rephrased taking into consideration this and other comments.

Page 3 line 7: What is the reason to choose 15 layers?

A- This is a compromise between representing important seasonal variations in temperature and (internal) mass balance, and keeping the computational cost to a minimum. This information has been added to the manuscript.

Page 3 line 9: To me it is not clear, how this follows from the previous sentences; maybe it is better without "thus".

A- We assume that this comments refers to line 9 on page 4 in the original manuscript. The logical connection is that the previous sentences explain how the adaptive grid mostly follows individual units of mass (Lagrangian). This sentence highlights that liquid water is an exception. However, this is not essential to the understanding and so we did remove the "thus".

Page 8 line 15: Please replace surface temperature by near surface temperature

**A- Done**

Page 12 line 15: The formulation "observed" could be misleading- maybe use something like "simulated" or "effective"

A- We now use the term "effective" throughout. Text and figures 5 and 6 have been revised.

Fig. 4: Does the scheme only transfer the mass balance to the ice model or also heat flux/temperature?

A- The ice sheet is a very simple vertically integrated model that at the moment does not use a temperature-dependent flow law. Therefore, energy fluxes are disregarded as they leave the lower domain boundary of the snow model.

Fig. 5 and 6: The model seems to be conserving mass and energy almost perfectly and I wonder if these figures could be reduced to fewer seasonal cycles, or even be replaced by some statistics, while the figures could be moved to the supplement.

A- For a model whose purpose is to provide the mass and energy balance as a boundary condition for another model, especially one that is intended to run over extended periods of time, we think it is essential to conserve these key properties and we would therefore prefer to keep the figures in the main text.

Page 13 Model calibration: The choice of calibration data sets should be motivated. I guess that the calibration is deliberately limited to data which are direct and relatively precise measurements (with the exception of the surface mass balance time series deduced from GRACE). However, the ablation zone is not well represented in the calibration and consequently, an evaluation of the spatial pattern of the SMB is important (see major comments).

A- For this comparison we are restricted to observations with reasonable temporal and spatial resolution and with reasonable accuracy. Of our prognostic and diagnostic variables, GrIS firn temperatures, seasonal snow coverage and the total mass balance fulfill these criteria the best and allow a calibration of both the mass and the energy balances. Analyzing the spatial pattern of the SMB is difficult not only because of the coarse resolution of BESSI but also because no direct observations exist with good spatial coverage of the Northern Hemisphere. We tried to accommodate for this by using the seasonal snow coverage, which represents the integrated effect of seasonal ablation. We prefer to not use data of other models such as RACMO in the calibration, partly because they only include a subset of our domain. However, we do now include a comparison in the discussion of the new figure 12. More details can be found in our reply to the major comments and to the comments on figure 11 below.

Page 17 lines 1-6 and Table 4: Here, the clarity could be improved. Maybe the parameter combination over the ten simulations with lowest RMSE could appear as TOP10x and together with BESTx, could be introduced in the text before line 4.

**A- This is a very good suggestion. Table 4 was changed.**

Page 17 lines 14-17: I don't understand, why higher wind speeds might reconcile relatively low optimal parameters of DSH. Also, at least over melting ice, wind speeds are rather reduced. And finally, the last sentence of this paragraph is not very clear to me.

A- We do not claim that low values are optimal for D\_sh but rather that this is a poorly constrained parameter. As a possible explanation, we speculate that since the bulk of the sensible heat exchange is by turbulent mixing, our choice to make D\_sh independent of wind speed is not ideal. This caveat repeated in the discussion. Please get back to us if this still does not clarify the issue, or if it does and specific changes should be made in the text.

Page 18 line 17-19: I don't find this analysis very convincing. I assume that surface temperatures are closely related to air temperature and even a PDD scheme would predict melt, if forced with daily temperatures > -50 C.

A- We agree that this analysis is somewhat qualitative and not a strong indication that the model performs well. For this reason we only show it after the objective ensemble calibration. We argue that we clearly describe the circumstances and caveats under which figure 9 was created.

Fig. 11a: What exactly is runoff? It does not seem to be runoff=rain+melting-refreezing. Generally, I don't seem to interpret Fig. 11 correctly. I don't see a good agreement with van den Broeke et al. (2016), who estimate SMB  $\approx$  300-400 Gt, accumulation is  $\approx$  600 Gt and refreezing is  $\approx$  200 GT.

A- We realize that our original statement about the good agreement with van den Broeke et al. (2016) was misleading in its brevity. In addition to that, the original figure contained two mistakes that we traced back to author AB misinterpreting model output that was originally generated by author MI. Thank you very much for catching these inconsistencies and we sincerely apologize for not being more thorough with our first submission.

BESSI agrees well with RACMO2.3 in the absolute values of melting and the resulting reduction in total mass balance. Interannual variations are captured in a similar way and the qualitative increase in refreezing and the snowfall being approximately constant also agree well. However, there are two mismatches that lead to the total mass balance in this model version being only 200 Gt/yr before the mid 1990s while RACMO2.3 simulates values around 400 Gt/yr. The first is a lower total precipitation in our forcing data from ERA interim as compared to RACMO. The differences accounts for approximately 100 Gt/yr. More importantly, BESSI underestimates the amount of refreezing by an additional 100-150 Gt/yr, which is substantial. This result is consistent with the comparison of BESSI with the model MAR in Plach et al. (2018), where refreezing was the most notable difference. The spatial pattern of the mismatch for present day climate reveals a meridional pattern in the mismatch (Fig. 7 in Plach et al., 2018). We speculate that this could be related to the lack of diurnal melt-freeze cycles as suggested by Dr. Krebs-Kanzow.

All of this information, the extended discussion, the corrected figure, and one new figure are included in the revised manuscript.

Page 20 line 2: Indeed, considering short wave radiation anomalies might be interesting. Is it possible to discuss this option?

A- This statement only refers to the ensemble with perturbed temperatures and precipitation. While it is in principle possible to also perturb shortwave radiation, we think that this should be part of a more comprehensive analysis. Given the length of the manuscript and the critical comment of reviewer Dr. Krapp on this section, we will not include additional experiments here. As mentioned above, a much expanded study with an updated code base, more than 15,000 simulations, and a robust statistical evaluation is close to completion.

**References**

Krapp, M., Robinson, A., and Ganopolski, A.: SEMIC: an efficient surface energy and mass balance model applied to the Greenland ice sheet, Cryosphere, 11, 1519–1535, https://doi.org/{10.5194/tc-11-1519-2017}, 2017.

Krebs-Kanzow, U., Gierz, P., and Lohmann, G.: Brief communication: An Ice surface melt scheme including the diurnal cycle of solar radiation, The Cryosphere Discussions, 2018, 1–11, https://doi.org/10.5194/tc-2018-130, https://www.the-cryosphere-discuss.net/tc-2018-130/, 2018.

Reeh, N.: Parameterization of melt rate and surface temperature on the Greenland ice sheet, Polarforschung, 59, 113–128, 1989.

Robinson, A., Calov, R., and Ganopolski, A.: An efficient regional energy-moisture balance model for simulation of the Greenland Ice Sheet response to climate change, The Cryosphere, 4, 129–144, https://doi.org/{https://doi.org/10.5194/tc-4-129-2010}, 2010.

van den Broeke, M. R., Enderlin, E. M., Howat, I. M., Kuipers Munneke, P., Noël, B. P. Y., van de Berg, W. J., van Meijgaard, E., and Wouters, B.: On the recent contribution of the Greenland ice sheet to sea level change, The Cryosphere, 10, 1933–1946, https://doi.org/10.5194/tc-10-1933-2016, https://www.the-cryosphere.net/10/1933/2016/, 2016.

**Referee #2, Mario Krapp**

First of all, I want to apologize to the authors and the editor for my late reply but some unforeseen circumstances have made a quicker reply impossible.

This paper by Born and colleagues describes a multi-layer layer snowpack and energy balance models which is suitable for large icesheet simulations on multi-millennial time scales. The snowpack model (BESSI) is part of the IceBern2D model and it has been tested with climatological data (ERA-Interim) in terms of its energy and mass conservation. The only shortcoming of this fast and efficient snowpack model is that it strongly depends on the value of atmospheric emissivity and the sensible heat ex change coefficient. The paper itself is well structured and has been written in a clear way.

This paper is a valuable contribution to the Cryosphere Community specifically for those interested in computationally efficient global snowpack models needed for interactive glacial–interglacial climate/icesheet model simulations of the past or large ensembles for future scenarios.

I would recommend this paper for publication in TC after the addressing the following remarks (my major point being the model/data availability, see below).

A- We are grateful for the time and effort that Dr. Krapp put into reviewing our work and in helping us improve it.

**Major Comments**

• My one major criticism of this paper is that the authors do not state if and how the model/ data will be made available (this affects how I scored the significance of this paper). The journal's data policy requires the following: "If the data are not publicly accessible, a detailed explanation of why this is the case is required". The authors mention that BESSI is part of IceBern2D but because IceBern2D does not seem to be publicly available (as far as I can tell), I suspect that neither will be BESSI. If this is really the case, than, in my point of view, this would be a bad decision in the light of scientifc transparency and reproducibility. The Cryosphere Community would defineltly benefit from making every model code and associated data publicly available (you can't simply reproduce such a complex model from scratch). As a best practise example, see Krapp et al. (2017). However, this is a decision to be made by the editor and/or in general by the editoral board of TC and not by me as a reviewer so I leave the final decision here to the editor. (I am sorry if this sounds a bit harsh but I strongly feel that we as a community need to be more transparent with what we do and show and I sense that Open Access is just one part of Open Science; for example: https://www.practicereproducibleresearch.org).

A- As we already suggested in our reply in the online discussion, we would like to publish the current code base as a supplement to this paper. Future versions, with development currently underway, may make use of a more sophisticated code-sharing and collaboration platform, but the current situation does not justify this additional effort.

• I find the title misleading. The authors do not show any glacial cycle simulation

A- We have changed the title to "An efficient surface energy-mass balance model for snow and ice".

• In the abstract you write ". . . even a marginal bias will develop into an erroneous solution over the long integration time and when amplified by strong positive feedback mechanisms": I am not sure that this is shown in the paper so please rephrase.

**A- This statement has been removed.**

• In the introduction, what is the benefit of having mutiple layers compared to a single layer? Is it possible to run BESSI in "single layer mode"; this would then allow a direct comparison of the effect of a single vs. multi-layer snowpack model.

A- Since a similar comment was made by reviewer Dr. Krebs-Kanzow, we refer to our reply above.

• Sect 2.1: As BESSI is integrated onto IceBern2D what is the reason that you didn't use the fully coupled version of IceBern2D/BESSI?

A- For the calibration with present-day data, uncertainties of the snow model should not be compounded with those of the ice sheet model. In addition, the simulations are not long enough to perturb the ice topography in a meaningful way.

• What about the contribution due to latent heat exchange in Eq (7)?

A- Latent heat exchange may occur in deeper layers and is therefore not included in equation 7. A short statement has been added to clarify this.

• The parameterizations of the different terms in the surface energy balance (Sec 2.3.1) indicate a few uncertainties: snow albedo (QSW), atmospheric emissivity (QLW), and wind speed and air pressure (QSH)

– What are the expected uncertainty ranges for each of those terms (as presented for QSH)

– E.g., atmospheric emissivity varies with cloud cover, snow albedo varies with liquid water content or dust particles, wind speed varies non-uniformly across an ice sheet, and air pressure changes with with ice sheet height

A- The original text included ranges for the albedo of dry and wet snow, atmospheric emissivity and wind speed from which uncertainty ranges for the individual heat fluxes can be derived. More detailed information such as regional variations and maps would require an extensive separate analysis that is too long to be included here. We apologize for promoting our future work once more, but this analysis is one of the main motivations for the ongoing study of Zolles et al.

• What are the vertical jumps in Fig. 5 a) and b) and 6 a)?

A- The abrupt changes are due to mass and energy transfer to the ice sheet model. This information has been added to the caption of figure 6. In figure 5 it was already included in the legend of panel b), so no further changes are necessary.

• p.19, l.1: In principle, BESSI could also be evaluated against snow temperature profile data from mountain glaciers, e.g., Gilbert et al. (2016), their Fig. 6 a–d and Fig. 9 (firnification data)

A- As mentioned in our reply to reviewer Dr. Krebs-Kanzow, we think the horizontal resolution of 40 km square is not good enough to meaningfully resolve mountain glaciers. BESSI is in principle capable of such simulations but if future applications want to use the model in this way, a separate calibration and evaluation will be necessary.

• I don't see the added value of Sect 5; it is rather confusing for the reader to start again with another model setup with a different climate model; I can't see why this section is important at all and thus this section fells short compared to the rest of the paper

A- Rather than only describing and calibrating the model, we think that an example of a use case is a valuable addition. An intriguing feature is the non-linear relationship between temperature and surface mass balance that was recently discussed with a highly simplified SMB model by Mikkelsen et al. (2018). We think that BESSI strikes an ideal balance between complexity and numerical efficiency to investigate this issue in more detail, and the analysis reveals that the conclusions of Mikkelsen et al. (2018) were too simplistic. This section can be ignored by readers without loss of consistency and so we prefer to keep it. We noticed a mistake in the labeling of figure 13 (now 14), which has now been corrected.

**Minor Comments**

• accumulation rate A and pressure change  $\Delta$  p should be added to Table 1

**A- Done**

• replace 273K with T0 = 273.15 K throughout the manuscript

**A- Done**

References

Krapp, M., Robinson, A., and Ganopolski, A.: SEMIC: an efficient surface energy and mass balance model applied to the Greenland ice sheet, The Cryosphere, 11, 1519-1535, https://doi.org/10.5194/tc-11-1519-2017, 2017.

Gilbert, A., Vincent, C., Six, D., Wagnon, P., Piard, L., and Ginot, P.: Modeling near-surface firn temperature in a cold accumulation zone (Col du Dôme, French Alps): from a physical to a semi-parameterized approach, The Cryosphere, 8, 689-703, https://doi.org/10.5194/tc-8-689-2014, 2014. Interactive comment on The Cryosphere Discuss., https://doi.org/10.5194/tc-2018-218, 2018.

---

## Author Response (AR2)

I believe you have done a very good job addressing the reviewers' comments. Furthermore, I find the manuscript of high quality and the model to be a valuable contribution to the field. However, there are still a few improvements that I believe should be made before publication, listed below.

P 7, line 18, Eq. 7 and subsections: A minor point, but consider ordering the terms homogeneously here in the text, in Eq. 7 and in the subsections beneath for simplicity and easier reference.

A- Done, including table 1.

Eq. 7: I agree with Mario Krapp concerning the lack of a latent heat term in this equation. Later it is stated that the latent heat of refreezing can only occur below the surface, however melting – and thus it would seem the latent heat of melting – can only happen at the surface. In other words, if $Q_{lh}$ (when negative) appears in the energy balance of Eq. 7 in the model, it should appear in the equation. And if this is not the case, this should be made much more clear in the description of how this is treated. Is Eq. 7 applied to the surface boundary of the first grid box, for example, while negative $Q_{lh}$ is applied within the box itself?

A- We have now included the latent heat term in equation 7, and moved the description from section 2.3.2 to 2.3.1. We also corrected a mistake in the description of the shortwave radiation, which is the net radiation, not downwelling.

P 12, line 20: These two sentences sound contradictory and is related to my previous comment: "Melting only occurs at the surface and the corresponding amount is added to water mass of the uppermost box. However, during one time step more than one snow layer may melt and the vertical grid is adjusted accordingly." Does melting only occur at the surface, or also within the snow layers – please clarify?

A- We have rephrased the second sentence.

P 13, line 1: After 5000 model years, the snow model is in equilibrium. => The snow model reaches equilibrium after 5000 model years. [Current formulation is a bit roundabout]

A- Done

Fig. 7 & Table 4: Figure 7 should come before Table 4, both in appearance and in the text. Fig 7 is more general and serves to introduce the individual metrics (A, T, and m), and these abbreviations should also be introduced in the text (around the paragraph on p 16, line 10).

A- We respectfully disagree that the order of figure and table needs to be changed. The abbreviations are now introduced in the text.

P 19, line 23-24: Surface melt should be a positive quantity.

A- This has been changed in the text but we prefer to keep melt as a negative quantity in corresponding figure 11.

P 20, line 15: RMSE with subscripts ($RMSE_A$, etc)?

A- This has been changed as suggested in the text and in figure 7.

P 20, line 17-18: I would move this first sentence to the end of the next paragraph, as the latter is more introductory to the section. Or generally modify this part. Otherwise it is missing a direct link into stating the BESSI experimental setup in the subsequent paragraph.

A- done

Fig. 5: The colors used are not particularly printer/reader friendly. At a minimum, please change the yellow to a darker variant. I would suggest using a colorblind safe palette, such as can be found here: http://colorbrewer2.org/, among others. This comment applies to other figures as well (mainly those with time series or lines).

A- The palette of figures 5, 11, and 13 has been changed. The green shade now contains a fair amount of blue to make it colorblind safe. We also corrected a minor inconsistency between figures 5 and 6 by interchanging the colors of the two curves in the latter.